# PTRAMP, CSS and Ripr form a conserved complex required for merozoite invasion of *Plasmodium* species into erythrocytes

Benjamin A. Seager[1,2], Pailene S. Lim [1,2], Xiao Xiao [1,2], Keng Heng Lai[3], Lionel Brice Feufack-Donfack[4], Sheena Dass[5], Nicolai C. Jung[1,2], Anju Abraham[1], Matthew J. Grigg [6,7], Nicholas M. Anstey[6,7], Timothy William[7,8], Jetsumon Sattabongkot [9], Andrew Leis [1,2], Rhea J. Longley [1,2,9], Manoj T. Duraisingh [5], Jean Popovici [4,10], Danny W. Wilson [3,11,12], Alan F. Cowman [1,2] ✉ & Stephen W. Scally [1,2] ✉

Invasion of erythrocytes by members of the *Plasmodium* genus is an essential step of the parasite lifecycle, orchestrated by numerous host-parasite interactions. In *P. falciparum* Rh5, with PfCyRPA, PfRipr, PfCSS, and PfPTRAMP, forms the essential PCRCR complex which binds basigin on the erythrocyte surface. Rh5 is restricted to *P. falciparum* and its close relatives; however, PTRAMP, CSS and Ripr orthologs are present across the *Plasmodium* genus. We investigated PTRAMP, CSS and Ripr orthologs from three species to elucidate common features of the complex. Like *P. falciparum*, PTRAMP and CSS form a disulfide-linked heterodimer in both *P. vivax* and *P. knowlesi* with all three species forming a complex with Ripr by binding its C-terminal region, termed the PTRAMP-CSS-Ripr (PCR) complex. Cross-reactive antibodies targeting the PCR complex differentially inhibit merozoite invasion. The crystal structure of a cross-reactive antibody reveals an inhibitory epitope on the C-terminal tail of PvRipr. Cryo-EM visualization of the *P. knowlesi* PCR complex confirms predicted models and demonstrates a core invasion scaffold in *Plasmodium* spp. with implications for vaccines targeting multiple species of malaria-causing parasites.

There are more than 200 species of *Plasmodium* that infect a diverse range of hosts including primates, rodents, reptiles and birds. At least six species, including *Plasmodium falciparum*, *P. vivax* and *P. knowlesi*, have the ability to infect humans. *P. falciparum* is the most lethal species to infect humans, while *P. vivax* is the most widespread globally[1,2]. *P. knowlesi* is confined mainly to regions of Southeast Asia and is transmitted from macaques by zoonotic infection[3,4]. The *Plasmodium* genus can be divided into three main subgenera or clades that

[1]The Walter and Eliza Hall Institute of Medical Research, Parkville, VIC, Australia. [2]University of Melbourne, Melbourne, VIC, Australia. [3]Research Centre for Infectious Diseases, School of Biological Sciences, The University of Adelaide, Adelaide, SA, Australia. [4]Malaria Research Unit, Institut Pasteur du Cambodge, Phnom Penh, Cambodia. [5]Department of Immunology and Infectious Diseases, Harvard T.H. Chan School of Public Health, Boston, MA, USA. [6]Global and Tropical Health Division, Menzies School of Health Research and Charles Darwin University, Darwin, NT, Australia. [7]Infectious Diseases Society Kota Kinabalu Sabah-Menzies School of Health Research Clinical Research Unit, Kota Kinabalu, Sabah, Malaysia. [8]Clinical Research Centre-Queen Elizabeth Hospital, Ministry of Health, Kota Kinabalu, Sabah, Malaysia. [9]Mahidol Vivax Research Unit, Faculty of Tropical Medicine, Mahidol University, Bangkok, Thailand. [10]Infectious Disease Epidemiology and Analytics G5 Unit, Institut Pasteur, Université Paris Cité, Paris, France. [11]Burnet Institute, Melbourne, VIC, Australia. [12]Institute for Photonics and Advanced Sensing (IPAS), University of Adelaide, Adelaide, SA, Australia. ✉e-mail: cowman@wehi.edu.au; scally.s@wehi.edu.au

include *Laverania* (includes *P. falciparum*), *Plasmodium* (includes most human infective species such as *P. vivax* and *P. knowlesi*) and *Vinckeia* (primarily rodent infective species).

The three clades of the *Plasmodium* genus represent distinct evolutionary branches distinguished by geographic distribution and severity of disease; however, they can also exhibit distinct host cell selectivity. *P. vivax* has a strict preference for invasion of reticulocytes in the blood whereas *P. falciparum* can invade both reticulocytes and the more mature normocytes[5]. While the core machinery for invasion of reticulocytes and normocytes by *P. vivax* and *P. falciparum*, such as the parasite actomyosin motor, is conserved, there are distinct ligand-receptor interactions that provide selectivity for host cell invasion (reviewed in ref. [6]). In *P. falciparum*, many of these ligands are dispensable for invasion[7]. The notable exception is reticulocyte-binding protein homolog 5 (Rh5)[8,9] which is an essential *P. falciparum* ligand that binds to the receptor basigin on human erythrocytes[10]. Rh5 can play a role in host tropism through polymorphisms in the protein and differential binding to basigin of other non-human primates[8,11].

In *P. falciparum* Rh5 functions in a complex of five proteins that include CyRPA (Cysteine Rich Protective Antigen)[12], Ripr (Rh5 interacting protein)[13], PTRAMP (*Plasmodium* thrombospondin-related apical merozoite protein)[14,15], and CSS (cysteine-rich, small, secreted)[16] that has been termed the PCRCR complex[17–19]. PTRAMP and CSS form a disulfide-linked heterodimer that tethers the PCRCR complex to the merozoite membrane via the transmembrane domain of PTRAMP[18]. All proteins in the PCRCR complex are functionally essential for *P. falciparum* merozoite invasion of human erythrocytes and, crucially, antibodies and nanobodies that bind to individual proteins can inhibit merozoite invasion[18,20–25]. Conditional gene knockouts of each PCRCR protein in *P. falciparum* display the same phenotype, whereby the merozoites can interact with the erythrocyte surface and produce the strong deformation of the host membrane typical during normal invasion; however, the merozoite fails to internalize[18,26]. The PCRCR complex has been hypothesized to capture and anchor the increased membrane surface contact formed between the merozoite and erythrocyte membrane that is created during strong deformation driven by the merozoite's actomyosin motor[18]. This facilitates the establishment of the moving junction and is followed by the downstream events of invasion and ultimately the internalization of the merozoite into the erythrocyte[27,28].

Despite being essential for *P. falciparum* invasion, Rh5 orthologs are absent in species outside of the *Laverania* subgenus[29,30]. Consequently, utilization of basigin as a host receptor for invasion is not universal, as demonstrated for both *P. knowlesi* and *P. vivax*[16], suggesting that other parasite ligand-receptor interactions facilitate host cell attachment in other *Plasmodium* spp. Non-*Laverania* species do, however, possess orthologs of other components of the *P. falciparum* PCRCR complex. *P. knowlesi* orthologs of PTRAMP, CSS and Ripr have been shown to be essential for merozoite invasion using conditional gene knockouts[16] and more recently a high-resolution transposon mutagenesis screen for growth[31,32]. It has been suggested that PkPTRAMP, PkCSS and PkRipr form a complex and that PkPTRAMP provides the means for erythrocyte binding[16]. The PkCyRPA ortholog was also identified, and while it has been shown to be essential for parasite growth, it does not appear to be part of this complex in *P. knowlesi*[16].

Here, we hypothesized that PTRAMP, CSS and Ripr form a common basis for invasion complexes across *Plasmodium* spp. We leveraged recent insights into the PCRCR complex to characterize these proteins in several species of *Plasmodium* to elucidate the conserved features of merozoite invasion complexes. Our findings revealed a conserved PCR trimeric complex common to all *Plasmodium* clades that forms a core invasion scaffold. Additionally, cross-reactive antibodies targeting PTRAMP, CSS, and Ripr were identified that differentially inhibit merozoite invasion of erythrocytes, including an antibody targeting Ripr that inhibited both *P. knowlesi* and *P.*

*falciparum* growth. Identification of a conserved molecular scaffold presents an attractive approach for the development of vaccines targeting multiple species of malaria-causing parasites.

## Results

### PTRAMP, CSS and Ripr are conserved in all clades of *Plasmodium*

We first confirmed that the proteins constituting the *P. falciparum* PCRCR complex are conserved in other *Plasmodium* spp. by searching for orthologs and found that PTRAMP, CSS and Ripr were present in all subgenera (Fig. 1a). This conservation contrasts with Rh5 which is only present in *P. falciparum* and other *Laverania* species. Whilst CyRPA is relatively conserved, none of the species within the *Vinckeia* subgenus possess a CyRPA ortholog. This suggests PTRAMP, CSS and Ripr (PCR) form a conserved three-membered complex present across all *Plasmodium* spp.

AlphaFold 3 was used to predict the structure of the PCR complex of different *Plasmodium* spp. to understand the assembly of the three proteins (Fig. 1b, Supplementary Fig. 1)[33,34]. This revealed a common architecture of the PCR complex in which PTRAMP and CSS together form a platform that is bound by the C-terminal end of Ripr. The PfCSS crystal structure aligned with the AlphaFold 3 prediction of PfPCR with an RMSD of 0.711 Å, suggesting that no conformational changes are required within PfCSS to facilitate PfRipr binding (Fig. 1b).

The engagement of Ripr with both PTRAMP and CSS in the AlphaFold 3 predictions was similar in all *Plasmodium* spp. but was inconsistent with existing biophysical data for *P. falciparum* which had suggested that PfCSS alone was sufficient for PfRipr binding[18]. However, this interaction was low affinity with the $K_D$ of CSS binding to Ripr in the low micromolar range[18,19]. In contrast, the PCR model predicted significant interaction of both PTRAMP and CSS with Ripr for *P. falciparum*, *P. vivax* and *P. knowlesi*. PfPTRAMP, PfCSS and PfRipr all contain multiple predicted N-linked glycan motifs and whilst there is very limited glycosylation in *P. falciparum* these proteins were expressed in insect cells and therefore are predicted to undergo extensive glycosylation (Fig. 1c)[35]. To further investigate PfPTRAMP-PfCSS (PfPC) - PfRipr binding, we expressed recombinant glycosylation-modified variants that either completely lacked glycans or had significantly reduced glycosylation (Fig. 1c, d).

Using biolayer interferometry (BLI), we determined that non-glycosylated forms of PfRipr and PfPC interacted with a $K_D$ of 158.5 nM (Fig. 1e), representing a ~10-fold stronger interaction compared to previous measurements (1–4 µM)[18,19] and equivalent to the affinity measured between Rh5 and CyRPA (179 nM)[36]. The enhanced binding affinity of non-glycosylated proteins suggests that N-linked glycans on PfRipr and PfPC, added during heterologous expression, had interfered with their interaction interface in previous studies. Despite removal of a majority of the PfPTRAMP glycans monomeric PfPTRAMP showed no binding to PfRipr (Supplementary Fig. 2). These results suggest that the complete PfPC heterodimer is necessary for high-affinity PfRipr binding.

### The *P. vivax* orthologs of PTRAMP and CSS form a disulfide-linked heterodimer

To investigate whether the disulfide-linked PC heterodimer is conserved as the basis for Ripr binding across *Plasmodium* species, as predicted by AlphaFold 3[34], we co-expressed *P. vivax* PTRAMP and CSS orthologs in mammalian cells. The resulting PvPC heterodimer could be separated into its component monomers through reduction of the intermolecular disulfide bond (Fig. 2a). Nanobodies were raised against the purified PvPC heterodimer to enable further structural and biophysical characterization (Supplementary Fig. 3). Crystallization of the PvPC heterodimer was achieved by truncating the predicted disordered N-terminal repeat region of PvCSS and adding nanobody D7 (Fig. 2b, Supplementary Tables 1–3). The resulting structure confirmed

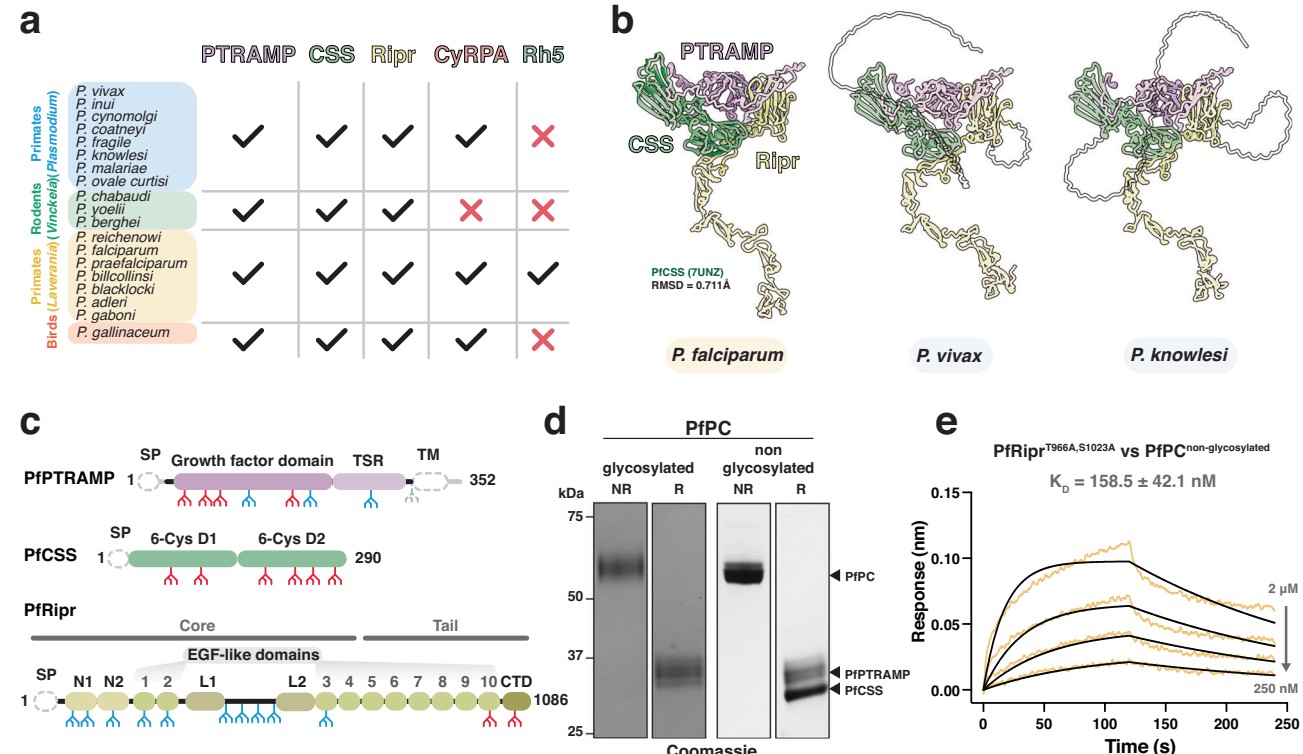

**Fig. 1 | PTRAMP, CSS, and Ripr are common to all clades of *Plasmodium*.**
**a** Comparison of PCRCR orthologs in the *Plasmodium* genus, showing that PTRAMP, CSS, and Ripr are common to all species, whereas Rh5 is restricted to the *Laverania* subgenus, and CyRPA is absent from the rodent-infective species (*Vinckeia*). **b** AlphaFold 3 predictions of PTRAMP, CSS, and the Ripr tail show a conserved architecture is predicted for each species. For the *P. falciparum* predicted complex, the previously published structure for PfCSS is superimposed in dark green. Regions that are predicted to be disordered, and therefore have low model confidence, are shown as transparent. The transmembrane domain and signal peptide have been removed from PTRAMP and CSS for clarity. **c** Domain diagrams for PfPTRAMP, PfCSS and PfRipr showing the predicted N-linked glycosylation sites[35] (blue) that were either mutated (red) or the sequon truncated (gray). Gray regions indicate the stretches of sequence either processed (signal peptides) or removed for recombinant expression, in the case of PfPTRAMP (transmembrane domain and cytoplasmic tail). **d** SDS-PAGE of glycosylated and non-glycosylated PfPC in both non-reduced (NR) and reduced (R) conditions. Purification was repeated at least three times with similar results. **e** A representative biolayer interferometry sensorgram of non-glycosylated PfRipr binding to non-glycosylated PfPC showing data (yellow) and 1:1 model best fit (black). Source data for all graphs and uncropped gels are provided as a Source Data file.

that PvCSS adopts the previously characterized two-domain degenerate 6-Cys fold seen in PfCSS (Fig. 2c, Supplementary Fig. 4)[18].

No electron density was observed for either the growth-factor domain (GFD) or the thrombospondin repeat (TSR) domain of PvPTRAMP, despite space being available within the crystal lattice (Supplementary Fig. 4c). This suggests that the majority of PvPTRAMP was insufficiently stabilized within the crystal lattice to produce coherent diffraction. Nevertheless, clear electron density extended from PvCSS cysteine 122, the predicted site of disulfide formation with PvPTRAMP (Fig. 2d). Modeling of PvPTRAMP residues 42 to 53 revealed the structural basis for PvPC heterodimerization (Fig. 2e, f). Specifically, PvPTRAMP forms an interrupted β-strand that extends across both β-sheets of the CSS D1 domain, establishing multiple backbone interactions (Fig. 2c, f, Supplementary Table 2).

Alignment of available PTRAMP and CSS sequences showed that the two cysteines involved in heterodimerization are conserved across most species (Supplementary Figs. 5, 6). One exception is *Plasmodium inui*, which has tyrosine and serine substitutions in PTRAMP and CSS, respectively (Supplementary Figs. 5, 6). Nevertheless, AlphaFold[33] modeling predicts a similar interface between PTRAMP and CSS in this region (Supplementary Fig. 1).

### PvRipr binds the PvPC heterodimer to form a high affinity complex

Biophysical analysis revealed that PvPC binds to PvRipr with high affinity ($K_D = 28.8 \pm 3.9$ nM) (Fig. 3a, Supplementary Fig. 7a, b). While monomeric PvPTRAMP was sufficient for binding, it showed approximately 10-fold weaker affinity ($K_D = 292.5 \pm 25.6$ nM) (Fig. 3b). No interaction was detected between PvRipr and PvCSS (Fig. 3c), and none of the PvPCR components bound to PvCyRPA at the tested concentrations (Fig. 3d). Mass photometry analysis confirmed the formation of a stable PvPCR complex, with PvPC and PvRipr each displaying monodisperse peaks when analyzed individually (Supplementary Fig. 7c). Formation of the PvPCR complex, following incubation of PvPC and PvRipr, was evidenced by the emergence of a higher molecular weight peak corresponding to a mass of $207 \pm 29$ kDa which is consistent with a 1:1:1 complex of PTRAMP:CSS:Ripr (Fig. 3e). The formation of a stable complex was further validated by the co-elution of PvPC and PvRipr in size-exclusion chromatography (Supplementary Fig. 7d).

### The C-terminus of Ripr is sufficient for PTRAMP-CSS binding in multiple species of *Plasmodium*

Previous studies have demonstrated that a three-membered PTRAMP-CSS-Ripr complex is involved in *P. knowlesi* invasion[16]. We confirmed that PkPTRAMP and PkCSS form a disulfide-linked heterodimer analogous to those observed in *P. falciparum* and *P. vivax* (Fig. 4a) and found that the PkPC heterodimer bound PkRipr with high affinity ($K_D = 0.6 \pm 0.1$ nM) (Fig. 4b, Supplementary Fig. 8a, b). Monomeric PkPTRAMP, but not monomeric PkCSS, was sufficient for this interaction (Fig. 4c, d). Like its *P. vivax* orthologs, PkPCR formed a stable complex as demonstrated by mass photometry, with a molecular

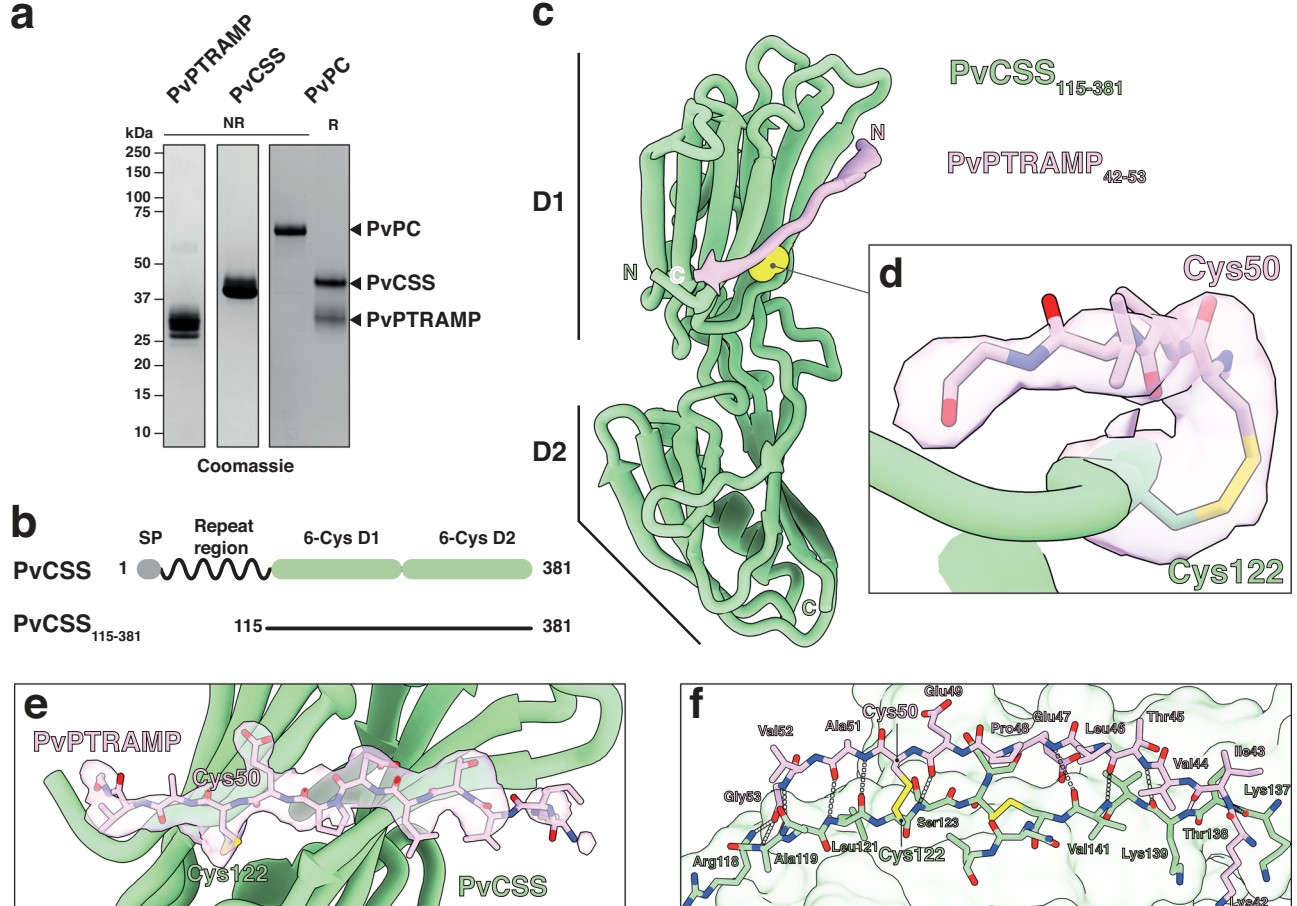

**Fig. 2 | Structure of the intermolecular disulfide bond between PvCSS and PvPTRAMP. a** SDS-PAGE of recombinantly expressed PvPTRAMP, PvCSS and PvPC heterodimer. Purification was repeated at least three times with similar results. **b** Domain diagram of PvCSS showing the disordered repeat region at the N-terminus. **c** Crystal structure of PvCSS$_{115-381}$(green) and PvPTRAMP$_{42-53}$(pink) with the disulfide formed between them in space-filling atomic depiction (yellow).

**d** Detail of the intermolecular disulfide bond between PvCSS and PvPTRAMP. Density is contoured at 1.0 σ and density extends to a range of 1.8 Å. **e** Structure of PvCSS and PvPTRAMP$_{42-53}$, showing the region of the unbiased electron density omit map that was attributed to PvPTRAMP. Density is represented as in **d. f** The interface between PvPTRAMP$_{42-53}$ and PvCSS. All intermolecular hydrogen bonds formed are represented in gray.

weight of 206 ± 13 kDa consistent with a 1:1:1 stoichiometry (Fig. 4e, Supplementary Fig. 8).

We performed a comparative biophysical analysis of PC-Ripr binding to identify the minimal regions of Ripr required for complex formation. Several truncations in the Ripr tail region were generated for *P. falciparum*, *P. vivax*, and *P. knowlesi* proteins and assessed for their ability to bind their cognate PC heterodimer (Supplementary Fig. 9)[19]. The tail region of Ripr, which encompasses epidermal growth factor (EGF)-like domains 5–10 and the C-terminal domain (CTD), was sufficient for heterodimer binding in all three species with no observable impact on affinity (Fig. 4f, Supplementary Fig. 9)[19]. A shorter construct containing only EGFs 9 and 10 plus the CTD also retained the ability to bind the heterodimer (Fig. 4f). Interestingly, complete removal of all EGF domains, leaving only the CTD, abolished binding for *P. falciparum* proteins but not for *P. vivax* or *P. knowlesi* (Fig. 4f). These results demonstrate that heterodimer binding requires only a discrete region of Ripr, with some species-specific differences in the minimal binding requirements.

### Anti-PCR antibodies are cross-reactive and differentially inhibit *Plasmodium* spp. invasion

Plasma samples from *P. falciparum*[37], *P. vivax*[38] and *P. knowlesi*[39,40] infected individuals from Thailand (Tha Song Yang) and Malaysia (Sabah) were assessed to determine the extent of patient antibody

response to the components of the PCR/PCRCR invasion complexes (Supplementary Fig. 10). IgG antibodies were assessed one week after clinical presentation and compared with malaria-naïve negative controls (IgG temporal kinetics from clinical presentation, one week, and one month post infection is shown in Supplementary Fig. 10). Significant IgG antibody reactivity was detected for *P. falciparum* patients against the PCRCR complex components except for PfCyRPA (Fig. 5a). Antibodies from *P. vivax* patients showed reactivity to the PvPCR components PvPC, PvCSS and PvRipr, but not PvPTRAMP (Fig. 5a). Similarly, antibodies from *P. knowlesi* patients showed reactivity to PkPC, PkCSS and PkRipr but not PkPTRAMP. Overall, there was a consistently low response to monomeric PTRAMP compared to other antigens and a consistently high response to monomeric CSS suggesting that the antibody response to the PC heterodimer is predominantly against CSS. High antibody responses were also observed for Ripr from all species and is consistent with Ripr being immunodominant as reported previously[41]. Furthermore, antibody responses to CSS and Ripr were broadly cross-reactive, with the antigens of all three species cross-reacting with sera from individuals independent of infective species (Supplementary Fig. 10).

We sought to investigate the potential of antibodies and nanobodies to inhibit growth of multiple *Plasmodium* species, given the serological cross-reactivity observed. Monoclonal antibodies (mAbs) and nanobodies generated against PvPC and mAbs against PvRipr were

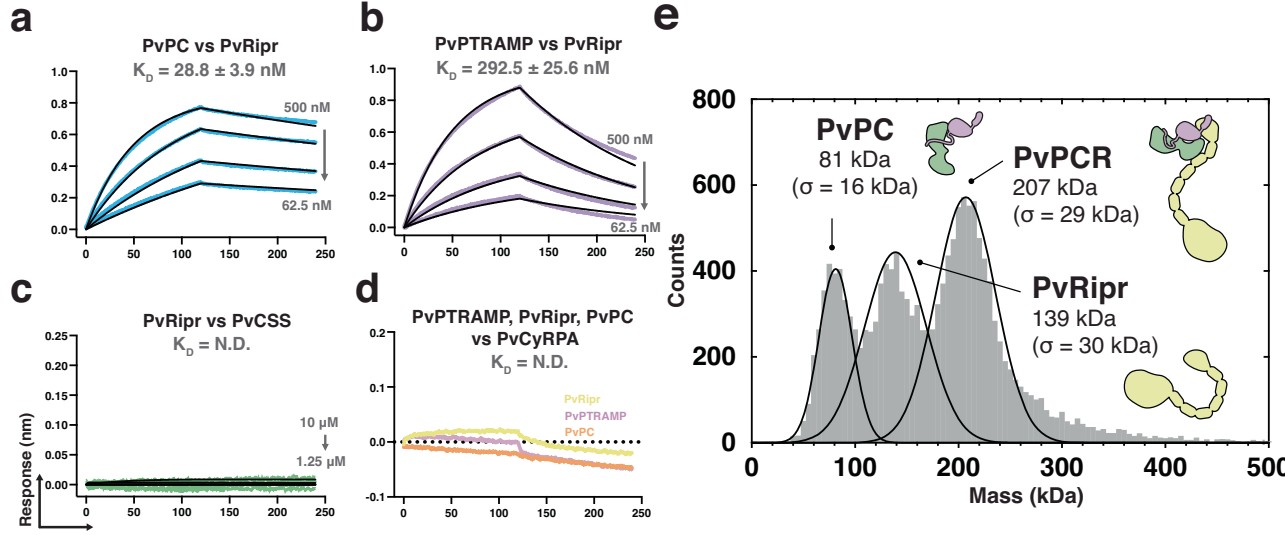

**Fig. 3 | The PvPC heterodimer forms a stable complex with PvRipr.**
**a–c** Representative biolayer interferometry sensorgrams of PvPC, PvPTRAMP, PvCSS, and PvRipr binding assays. Dilution series data are shown in color and 1:1 model best fit is shown in black. **d** Representative sensorgrams of PvPTRAMP, PvRipr and PvPC binding to PvCyRPA at an analyte concentration of 5 μM. Data could not be reliably fit with a model, and so no best fit has been shown. **e** Mass distribution of PvPC and PvRipr after pre-incubation as measured by mass photometry. Histogram data are shown in gray and Gaussian curve fit in black. Source data for all graphs are provided as a Source Data file.

evaluated for cross-reactivity with their *P. falciparum* and *P. knowlesi* orthologs (Supplementary Figs. 11–13). All tested antibodies bound to PkPC and PkRipr, with four out of seven showing cross-reactivity across all three species (Fig. 5b, Supplementary Figs. 11, 12)[42]. All anti-PvPC nanobodies were cross-reactive with PkPC, however, this cross-reactivity was much lower against PfPC with only one out of eight binding PfPC (Supplementary Fig. 13). Epitope binning revealed that both 2D9 and 4E2, which bind PvPC, compete for Ripr binding. Of all the Ripr binding antibodies, only 4H10, which binds the tail of Ripr excluding EGF 6–8, competes for PvPC binding. Notably, anti-Ripr mAbs 5B3 and 5B4 both recognize EGF 6–8, a region previously associated with inhibitory activity in *P. falciparum* (Supplementary Fig. 12d)[41,43]. Growth inhibition assays (GIAs) were performed against *P. knowlesi* to assess the inhibitory potential of anti-PvPC nanobodies and anti-PvPCR antibodies. Initial screening revealed that two anti-Ripr antibodies (5B3 and 5B4) and both anti-PC antibodies (2D9 and 4E2) inhibited parasite growth at 0.5 mg/mL (Fig. 5c). This inhibition was dose-dependent, with 5B3 and 5B4 showing half-maximal effective concentrations ($EC_{50}$) of 74 μg/mL and 520 μg/mL respectively, while 2D9 and 4E2 exhibited an $EC_{50}$ of 635 μg/mL and 892 μg/mL respectively (Fig. 5d). The cross-reactive mAb 5B3 also inhibited *P. falciparum* growth, albeit with a significantly higher $EC_{50}$ of approximately 2 mg/ mL (Fig. 5e). In contrast, neither 4E2 nor 2D9 inhibited *P. falciparum* growth, despite showing moderate inhibition in *P. knowlesi*. This reduced inhibitory effect may be attributed to varying affinities of these antibodies for different Ripr orthologs (Supplementary Fig. 12e). The mAb 4H10 showed no inhibitory activity in *P. knowlesi* or *P. falciparum*[18].

Following screening of the antibodies in *P. knowlesi* and *P. falciparum* GIAs (Fig. 5c–e, Supplementary Fig. 14), we assessed their potential inhibitory effect on *P. vivax* merozoite invasion and parasite growth in ex vivo GIAs. Assays performed on Cambodian *P. vivax* parasites revealed no inhibitory effect for any of the tested antibodies (Fig. 5f). To validate the *P. vivax* results, we evaluated a subset of these antibodies for their ability to inhibit growth in the closely related species *P. cynomolgi*[44]. The data from these assays were consistent with the *P. vivax* findings, confirming that none of the antibodies could inhibit parasite growth in either of these two species (Fig. 5g). These results demonstrate that while antibodies against the PCR complex may exhibit cross-reactivity across recombinant PCR complexes from multiple species of *Plasmodium*, this cross-reactivity does not necessarily correlate with growth inhibitory capacity. As these antibodies were raised against the *P. vivax* protein, these results either suggest minor functional differences between the complexes of *P. falciparum* and *P. knowlesi* compared to *P. vivax* and *P. cynomolgi* or that the PCR components are less critical for invasion of *P. vivax* and *P. cynomolgi*.

### The anti-PvRipr antibody 5B3 targets a conserved epitope on EGF 7 and 8 of Ripr

To understand the molecular basis of the cross-reactivity exhibited by the inhibitory antibody 5B3, we determined the crystal structure of the antigen-binding fragment (Fab) of 5B3 in complex with PvRipr^EGF 7–8 (Fig. 6, Supplementary Table 4). Both EGF 7 and EGF 8 adopt canonical EGF-like folds, comprising two antiparallel β-sheets and three surface-exposed loops. The domains are connected via a short EGF 7 linker (residues Phe801–Asn803)[45–47]. 5B3 primarily engages the β-sheets, B-loop, and C-loop of EGF 7, with additional contacts to the EGF 7 linker and the $3_{10}$-helix of EGF 8 (Fig. 6a). Sequence alignment of Ripr EGF 7–8 domains from *P. vivax*, *P. knowlesi*, *P. falciparum* and *P. cynomolgi* orthologs revealed 71% sequence similarity. 79% of the total surface area on PvRipr^EGF 7–8 buried by antibody 5B3 is conserved across species, providing a structural explanation for its cross-species reactivity. A small number of polymorphic residues, such as Glu786 and Asn802, contribute to hydrogen bonds or salt bridges but make only moderate contributions to the total buried surface area (BSA) (Fig. 6b, Supplementary Table 5). The antibody-antigen interface is dominated by heavy chain contacts, with interactions mediated by all three heavy complementarity-determining regions (HCDRs) and framework region 3 (HFR3). In contrast, the light chain contributes only a single contacting residue, Trp91, from LCDR3. Most interactions involve conserved epitope residues (Fig. 6c, Supplementary Table 5), consistent with 5B3's cross-reactivity. Among the few polymorphic contact residues Glu786 in PvRipr forms both a salt bridge and a hydrogen bond with H-Arg73, as well as an additional hydrogen bond with H-Ser28 (Fig. 6d). Substitution with valine in PfRipr would abolish those specific interactions, yet all van der Waals interactions with HCDR1 and HFR3

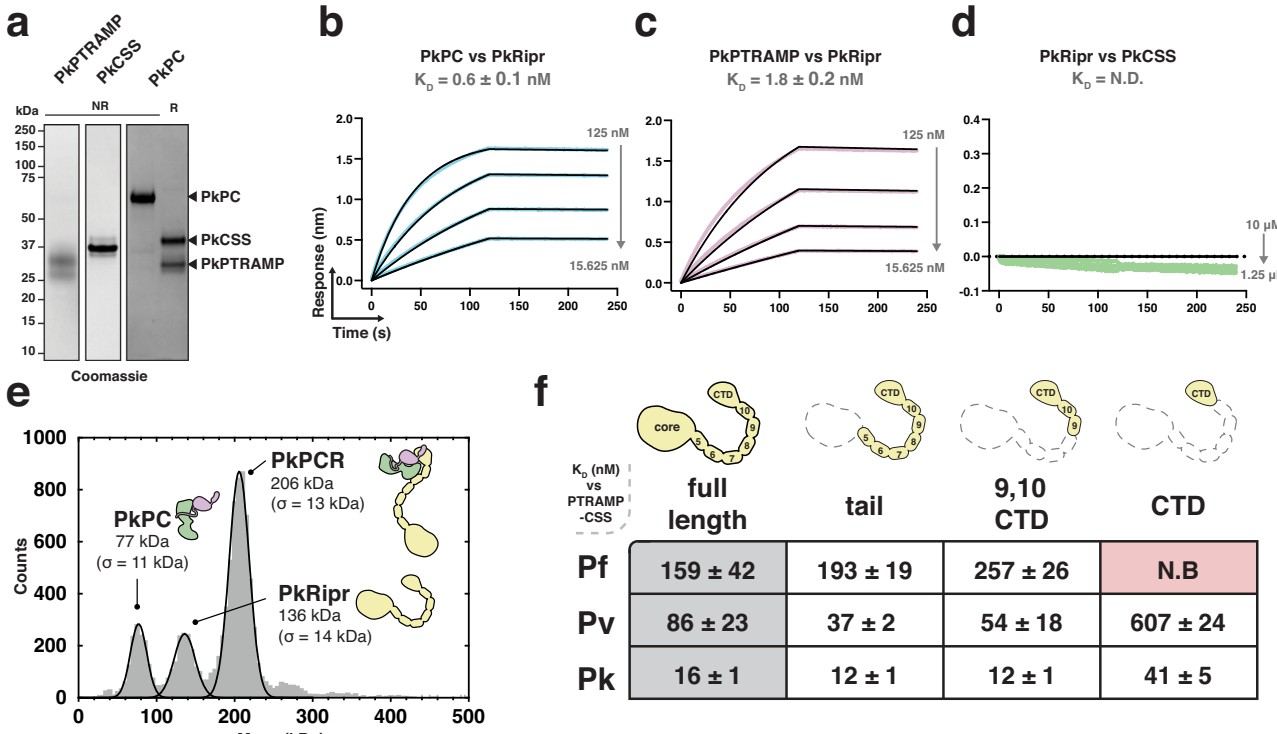

**Fig. 4 | A small region of Ripr is sufficient for PTRAMP-CSS binding in *P. falciparum*, *P. vivax*, and *P. knowlesi*. a** SDS-PAGE of recombinantly expressed PkPTRAMP, PkCSS and PkPC heterodimer. Purification was repeated at least three times with similar results. **b–d** Representative biolayer interferometry sensorgrams of PkPC, PkPTRAMP, PkCSS and PkRipr binding assays. Dilution series data are shown in color and 1:1 model best fit is shown in black. **e** Mass distribution of PkPC and PkRipr after pre-incubation as measured by mass photometry. Histogram data are shown in gray and Gaussian curve fit in black. **f** The ability of Ripr and Ripr truncations to bind to PTRAMP-CSS in *P. falciparum*, *P. vivax* and *P. knowlesi*. The table shows the truncations used and their dissociation constant ($K_D$, in nM, with standard error of the mean (SEM)) for binding their cognate heterodimer. N.B indicates no binding. Source data for all graphs are provided as a Source Data file.

are predicted to be preserved. Similarly, Asn802 interacts with multiple tyrosine residues in HCDR1 and HCDR3. Structural modeling suggests that the Asn802Glu substitution in PfRipr is well tolerated owing to the similar side chain size, with sufficient spatial accommodation and the potential to form additional hydrogen bonds within the tyrosine-rich environment, without disrupting existing interactions (Fig. 6e). Together, these data reveal that 5B3 engages a highly conserved epitope across Ripr orthologs. While a limited number of polymorphic Ripr residues contribute to the interface, they do not abrogate binding, supporting 5B3's broad reactivity against Ripr from multiple *Plasmodium* species.

### Cryo-EM analysis of PkPCR supports AlphaFold predictions

To provide more confidence in the predicted models of the PCR complexes, and to understand how inhibitory antibodies may function, we carried out cryo-electron microscopy (cryo-EM) experiments on the PkPCR complex. Cryo-EM analysis of the PkPCR$^{tail}$ complex revealed an overall shape consistent with the AlphaFold prediction (Fig. 7a, Supplementary Fig. 15). The addition of 5B3 Fab allowed unambiguous assignment of the orientation of the two-dimensional (2D) classes (Fig. 7a). Furthermore, comparison of Fab bound and unbound classes showed no discernible differences in the PCR complex which confirmed that 5B3 binds to the tail region of Ripr without interfering with complex formation (Fig. 5b, Supplementary Figs. 12, 15). This suggests that parasite inhibition by 5B3 likely has a direct effect on Ripr function during invasion rather than on the complex. The cryo-EM data, combined with the PvPC crystal structure, strongly support the predicted PCR complex structure (Fig. 7b). In this complex, the PTRAMP-CSS heterodimer is formed by an intermolecular disulfide bond. This heterodimer engages Ripr via two interfaces:

PTRAMP clinching the CTD of Ripr, and the D2 domain of CSS interacting with EGF 9 of Ripr (Fig. 7b). The PTRAMP-Ripr interface is predicted to involve several complementary electrostatic interactions between the CTD of Ripr with both the GFD and TSR of PTRAMP. The CSS-Ripr interface is predicted to be predominantly hydrophobic in nature, with the two β-strands of EGF 9 packing against the α-helix of CSS and its neighboring β-strand. In total, the predicted interface formed between PkRipr and the PkPC heterodimer has a BSA of approximately 2,000 Å². The remaining mass of Ripr likely extends below the PCR complex, where it may interact with other invasion proteins.

While PfPC lacks the ability to bind erythrocytes directly, it enhances Rh5 binding when incorporated into the PCRCR complex[18]. Previous studies have shown that PkPTRAMP can bind erythrocytes; however, these experiments were performed with monomeric PkPTRAMP and not heterodimeric PkPC[16]. We performed flow-cytometry based erythrocyte binding assays to assess whether PkPC or the PkPCR complex bound to erythrocytes. Neither PkPC nor PkPCR showed significant binding to erythrocytes relative to the positive control, PfRh5 (Fig. 7c, Supplementary Fig. 16a). Considering that *P. vivax* invades reticulocytes exclusively, we extended our investigation to include reticulocyte binding assays. Similarly, we observed no binding of PvPC or PkPC to reticulocytes (Fig. 7d, Supplementary Fig. 16b). Collectively, our analysis found no evidence of erythrocyte or reticulocyte binding by PkPC, PvPC, or the PkPCR complex.

We therefore hypothesize that PTRAMP, CSS and Ripr form a core invasion scaffold in *Plasmodium* parasites. This scaffold provides the basis for the assembly of species-specific complexes that are adapted for binding a diverse set of host erythrocyte receptors (Fig. 7e). In *P. falciparum* this complex incorporates CyRPA and Rh5 which facilitate

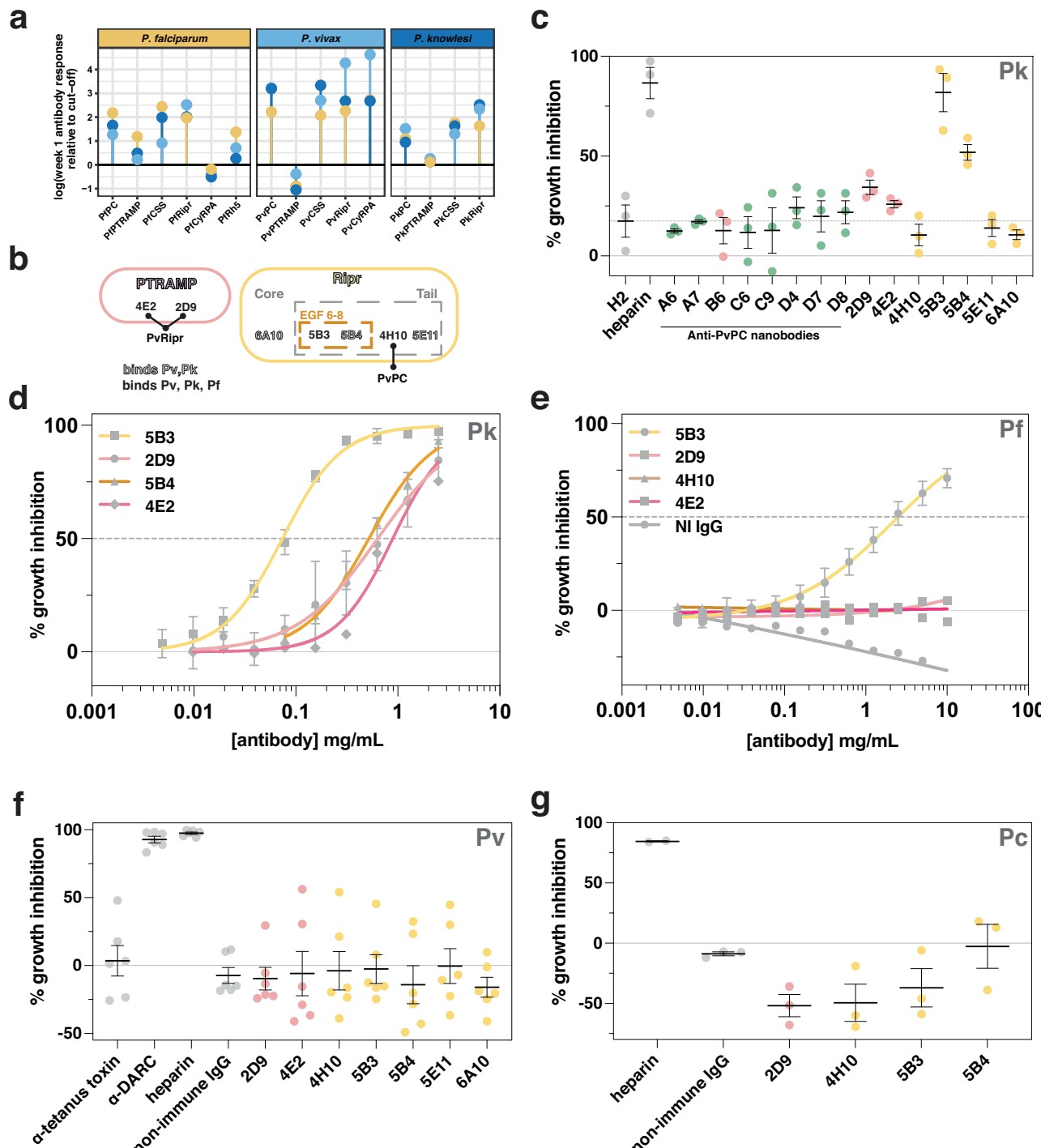

**Fig. 5 | Cross-reactive PTRAMP, CSS, and Ripr antibodies exhibit differential inhibition in multiple species of *Plasmodium*. a** IgG antibodies in human patients with *Plasmodium* infections. Fold-change peak week one antibody response relative to the seropositivity cut-off (mean of negative controls + 2x standard deviation). **b** Mouse monoclonal antibodies mapped by binding region. Open text represents antibodies binding both *P. vivax* and *P. knowlesi*, and closed text represents cross-reactivity between all three species. Bold lines indicate 4E2 and 2D9 block PvPC binding to PvRipr and that 4H10 blocks PvRipr binding to PvPC. All other antibodies have no effect on PvPC-PvRipr binding and are assumed to be capable of binding the PvPCR complex. **c** *P. knowlesi* growth inhibition assay (GIA) of anti-PvPC and anti-PvRipr biologics. The non-inhibitory PfCSS nanobody H2 was included as a negative control[18]. Three independent experiments were performed with mean and SEM shown. Antibodies and nanobodies were tested at a final concentration of 0.5 mg/mL. Data are colored according to antigen (CSS in green, PTRAMP in pink and Ripr in yellow). **d** GIA dilution series for inhibitory antibodies

in *P. knowlesi*. Growth inhibition (%) is the mean of six independent experiments for 5B3, 2D9, four independent experiments for 5B4 and two independent experiments for 4E2. Error bars represent standard deviation. **e** GIA dilution series for inhibitory antibodies in *P. falciparum*. Growth inhibition (%) is the mean of five independent experiments for 5B3, two independent experiments for 4E2, and one independent experiment for 4H10, 2D9 and non-immune IgG. Error bars represent standard deviation. **f** Ex vivo GIA of *P. vivax* parasites. Antibodies were tested at a final concentration of 0.5 mg/mL. Anti-Duffy antigen receptor for chemokines (DARC) mouse monoclonal antibody 2C3 was used as a positive control[72]. Data are from six independent experiments. Error bars show mean and SEM. Data are colored as in **c**. **g** *P. cynomolgi* GIA of anti-PvPC and anti-PvRipr antibodies. Antibodies were tested at a final concentration of 0.5 mg/mL. Three independent experiments were performed with mean and SEM shown. Data are colored as in c. Source data are provided as a Source Data file.

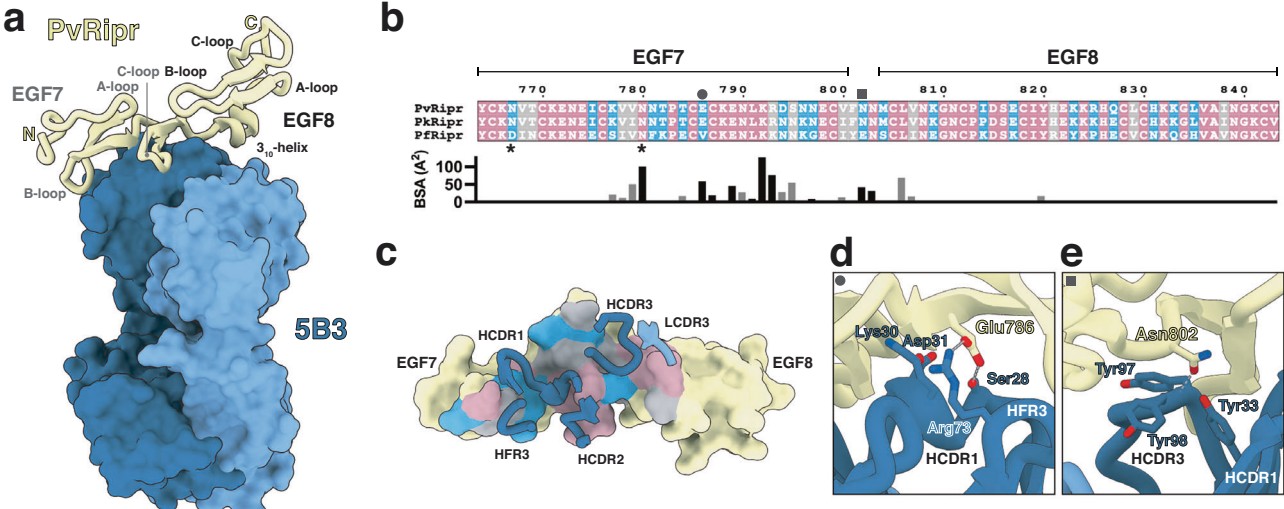

**Fig. 6 | Crystal structure of anti-Ripr mAb 5B3 in complex with PvRipr^EGF7-8.**
**a** Crystal structure of PvRipr^EGF7-8 (yellow) in complex with 5B3 Fab, with heavy chain in dark blue and light chain in light blue. **b** Sequence alignment of the EGF 7-8 domains from PvRipr, PkRipr, PcRipr and PfRipr. Similarity was predicted using ESPript 3.0 with the "% equivalent" parameter scheme. Identical residues are shown in pink, residues with similar physicochemical properties are in gray and residues with distinct properties in cyan. Mutated N-linked glycosylation sites (Asn767Gln and Asn780Gln) are marked with asterisks. Below is the surface area contribution of each PvRipr EGF 7-8 residue buried by antibody 5B3 as determined by PISA. Residues forming hydrogen bonds of salt bridges with 5B3 are colored black; those involved in van der Waals interactions are gray. **c** Surface representation of the PvRipr^EGF7-8 with the 5B3 epitope colored according to sequence similarity as in **b**. **d, e** Detailed views of interactions between 5B3 and polymorphic Ripr residues Glu786 in **d** and Asn802 in **e**. Hydrogen bonds and salt bridges are shown as black dashed lines.

invasion via the essential interaction with basigin. The equivalent proteins in *P. vivax* and *P. knowlesi* that are responsible for erythrocyte binding are yet to be identified. If findings from *P. falciparum* are applicable to these species, these interactions will be an essential step in merozoite invasion.

## Discussion

The highly conserved nature of PTRAMP, CSS and Ripr across the *Plasmodium* genus, combined with their demonstrated essential roles in both *P. falciparum*[18] and *P. knowlesi*[16] invasion, positions these proteins as compelling targets for understanding fundamental mechanisms of merozoite invasion. Our cross-species structural and functional analyses, enabled by recent advances in protein structure prediction through AlphaFold[33], reveal that PTRAMP, CSS and Ripr form a conserved invasion scaffold in *Plasmodium* parasites. This scaffold serves as a foundation for the assembly of species-specific protein complexes (Fig. 7e), highlighting a delicate balance between conservation and adaptation in *Plasmodium* invasion mechanisms.

Structural analysis of PvPC revealed a critical intermolecular disulfide bond between PvPTRAMP and PvCSS. The essentiality of this linkage was previously established in *P. falciparum* invasion[18], and the evolutionary conservation of these cysteine residues across *Plasmodium* species strongly suggests that PTRAMP-CSS heterodimerization represents a fundamental feature throughout the genus. Optimization of recombinant heterodimeric PfPC revealed a much tighter interaction with PfRipr than previously reported[18,19], aligning with the structural architecture predicted by AlphaFold[33]. Our biochemical studies demonstrated that the heterodimeric PkPC forms a high-affinity complex with PkRipr, corroborating earlier pull-down mass spectrometry data from *P. knowlesi* parasites[16] and providing robust evidence for the biological significance of this complex in vivo. The formation of this trimeric complex extends beyond *P. knowlesi* and *P. falciparum*, as we also demonstrated its assembly in *P. vivax*, providing evidence for a conserved molecular feature across multiple *Plasmodium* species.

We have refined the understanding of Ripr's role in complex formation, showing that only a small domain within the Ripr tail is required for PC heterodimer binding. In *P. falciparum*, this binding region encompasses EGFs 9 and 10 and the CTD of PfRipr, while in *P. vivax* and *P. knowlesi*, the CTD alone is sufficient. The considerable length of Ripr (>150 Å) in *Plasmodium* species may enable the PCR/PCRCR complex to bridge the gap between the merozoite surface and host cell membrane during invasion (Fig. 7e)[19]. This hypothesis is supported by previous studies showing that antibodies targeting EGF domains 6, 7, and 8 within the Ripr tail effectively inhibit parasite growth in vitro[24,41].

Our analysis of antibodies from infected patients revealed significant cross-reactivity to CSS and Ripr across *Plasmodium* species, raising the potential for cross-species antibody-mediated inhibition of invasion. It is unlikely that these patients had previously been recently infected with all three *Plasmodium* species, particularly given the low transmission in these settings, suggesting that antibodies generated against the PCR complex targeted conserved epitopes. The identification of cross-species neutralizing antibodies is an attractive finding for vaccinology[45,46], and previously anti-CelTOS monoclonal antibodies have been identified that have cross-species activity against *P. falciparum* and *P. vivax* as well as demonstrated multistage activity against liver infection and preventing transmission to mosquitoes[48]. The conservation of the PCR complex makes it an attractive target for a similar approach.

Interestingly, the Ripr-binding mAb 5B3, raised against PvRipr, exhibited an unexpected pattern of cross-inhibitory activity. It inhibited invasion of *P. knowlesi* and *P. falciparum*, but not *P. vivax* or *P. cynomolgi*, despite binding to the PvPCR complex. This paradoxical result suggests that the mechanism of inhibition is more complex than simple binding, and that accessibility or functional importance of certain epitopes may differ between species during invasion[49]. These findings highlight intricate species-specific functional adaptations that are not immediately apparent from structural data alone. The PCR complex likely interacts with other invasion proteins that are not conserved across species. For instance, in *P. falciparum*, this complex includes CyRPA and Rh5, which mediate the essential interaction with basigin[10]. While equivalent proteins to Rh5 in *P. vivax* and *P. knowlesi*

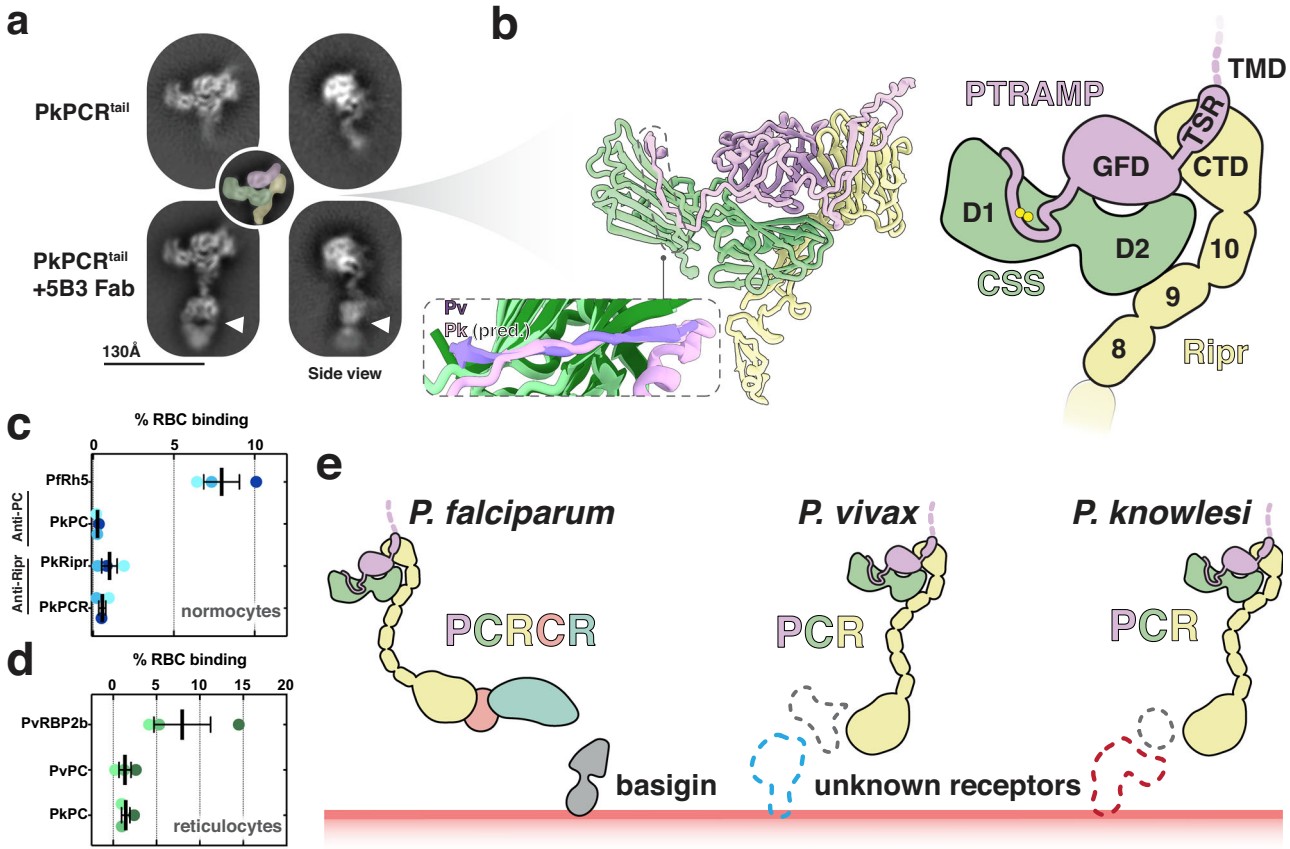

**Fig. 7 | PTRAMP, CSS, and Ripr form a core invasion scaffold in *Plasmodium* spp. a** Cryo-EM 2D class averages of the PkPCR<sup>tail</sup> complex. Addition of the 5B3 Fab fragment shows distinct density at the end of the Ripr tail (white triangle). Inset shows the PkPCR<sup>tail</sup> complex colored according to the predicted structure. **b** AlphaFold 3 predicted structure (left) and cartoon schematic (right) of PkPCR<sup>tail</sup>. Transmembrane domains, signal sequence and large disordered regions have been omitted for clarity. Inset shows the alignment of the PvPC (purple and dark green) structure and the PkPC predicted structure (pink and light green). **c** Erythrocyte binding assays of PkPC, PkRipr and PkPCR. Each color corresponds to one erythrocyte donor. PfRh5 was used as a positive control. Data from three independent experiments with three individual donors are shown with mean and

SEM. **d** Erythrocyte binding assays of PkPC and PvPC using reticulocyte enriched cord blood. PvRBP2b was used as a positive control. Data from three independent experiments with three individual donors are shown with mean and SEM. **e** Schematic of the PCRCR complex of *P. falciparum* and PCR complexes of *P. vivax*, and *P. knowlesi*. PTRAMP, CSS and Ripr form a conserved three-membered complex that serves as a scaffold for an invasion complex. In *P. falciparum* this complex involves CyRPA and Rh5. In *P. vivax* and *P. knowlesi* this complex likely contains other components (dashed gray) that have yet to be identified, which engage with the erythrocyte membrane via host-cell specific receptors. Source data are provided as a Source Data file.

remain unidentified, the conservation of the core scaffold suggests analogous receptor-ligand interactions govern invasion in these species. The functional divergence reflects evolutionary adaptations to different host cell environments[29] and species-specific differences in conformational flexibility. These dynamic properties, not captured in static structural models, could contribute significantly to functional divergence.

The ability of some anti-PvPC antibodies to inhibit parasite invasion, despite blocking PCR complex formation, challenges existing data on PCRCR complex pre-formation in micronemes. This suggests a need to reconsider PCR/PCRCR complex function and points to previously unappreciated complexity in the invasion process. Thus, it is possible that there is some complex formation following microneme discharge. Or that structural flexibility within this region allows antibody binding that would otherwise be sterically hindered by members of the complex, similar to how some potent anti-Pfs48/45 mAbs can recognize epitopes in the context of Pfs230[50]. To fully elucidate the functional divergence across *Plasmodium* species, further studies are needed[6]. These should include structural analyses of complete

invasion complexes from multiple species and detailed investigation of complex dynamics during the invasion process. Additionally, exploring the interaction of the PCR complex with species-specific partners and host cell receptors will be crucial in building a comprehensive picture of invasion mechanisms across the genus.

In conclusion, this interplay between structural conservation and functional adaptation highlights the complexity of *Plasmodium* invasion mechanisms. It underscores the importance of comparative studies across species and emphasizes the need for caution when extrapolating functional insights from one *Plasmodium* species to another, despite apparent structural similarities. These findings not only advance our understanding of malaria parasite biology but also have important implications for the development of cross-species interventions against this devastating disease.

## Methods
### Recombinant protein expression
All gene sequences used were retrieved from the VEuPathDB (accessed through www.plasmodb.org)[51] from reference strains (3D7 for *P.*

*falciparum*, PvP01 for *P. vivax* (with the exception of PvRBP2b for which the *Sal*-1 sequence was used), and strain H for *P. knowlesi*). All genes were synthesized by Genscript (Singapore) unless otherwise stated.

## P. falciparum

Recombinant PfRh5, PfCyRPA, PfPTRAMP, PfPC and PfRipr constructs were produced as described previously, with some changes made to the synthetic gene constructs used for expression[18]. PfPTRAMP, comprising residues 31 to 307, was subcloned into the pAcGP67a vector with a C-terminal C-tag. Four potential N-linked glycosylation sites were removed, at positions Asn112, Asn149 and Asn155 by mutation to Gln, and at position Asn195 by mutation of Thr197 to Ala, to produce PfPTRAMP_31-307_4x. To produce a large amount of pure monomeric PfPTRAMP, another construct was made that contains all the same mutations and also contains Cys60Ser mutation to prevent disulfide formation, termed PfPTRAMP_31-307_4xC60S. PfCSS was subcloned into the pAcGP67a vector with a C-terminal FLAG-tag preceded by a TEV protease cleavage site. This construct has all six potential N-linked glycosylation sites removed at positions Asn74, Asn88, Asn192, Asn234, Asn261 and Asn283 by mutation of Ser76, Thr90, Ser194, Thr236 and Thr263 to Ala and Asn283 to Gln, to produce PfCSS_21-290_6×[18]. The previous PfRipr construct[18] was altered with the following mutations: Thr966Ala-Ser1023Ala yielding the construct PfRipr_20-1086_2xA. PfRipr$^{tail}$(aa 717-1086), PfRipr$^{EGF\ 9,10,CTD}$(aa 899-1086) and PfRipr$^{CTD}$(aa 981-1086), were all synthesized by Genscript and purified in an identical fashion to PfRipr.

## P. vivax

The *pvptramp* gene (PVP01_1436800, aa 21-297) excluding the transmembrane and cytoplasmic domains was subcloned into a modified pTRIEX2 vector that contains an N-terminal Small Ubiquitin-like Modifier (SUMO)-Flag tag followed by a Tobacco Etch Virus (TEV) protease cleavage site (from here on termed SFT). Potential N-glycosylation sites were assessed and one site, Asn115, was removed by mutation of Ser117 to Ala. This yielded SFT_PvPTRAMP_21-297_S117A. This construct was then further cloned to incorporate a C-terminal Avitag, yielding SFT_PvPTRAMP_21-297_S117A_Avi. Both constructs were expressed in Human Embryonic Kidney (HEK) Expi293F cells (Life Technologies) as secreted soluble proteins. Transient transfection was carried out as per the manufacturer's protocol and the culture medium harvested 5–6 days post-transfection. The proteins were purified via multiple rounds of binding and eluting using Anti-Flag M2 Affinity Gel (Merck) and 100 μg/mL of Flag peptide (Genscript) in HBS (20 mM HEPES pH 7.2, 150 mM NaCl). The eluted fractions were pooled and incubated with TEV protease (1 mg of TEV for every 10 mg of protein) overnight at 4 °C. His-tagged TEV was removed by applying the protein solution to nickel-nitrilotriacetic acid (Ni-NTA) agarose resin (Qiagen) and collecting the flowthrough. The flowthrough was then concentrated on a 10,000 dalton (Da) Molecular Weight Cut-Off (MWCO) Amicon Ultra-15 Centrifugal Filter (Merck) and applied to an S75 Increase 10/300 column (Cytiva) connected to an Akta Pure (Cytiva) to separate the TEV-cleaved PvPTRAMP from the SUMO-Flag tag. Peak fractions were assessed for purity via sodium dodecyl sulfate–polyacrylamide gel electrophoresis (SDS-PAGE) and pure fractions were pooled and concentrated. Glycerol was added (10% v/v final) to the concentrated protein and then aliquoted and flash frozen in liquid nitrogen (LN$_2$).

The full-length *pvcss* gene (PVP01_1344100, aa 22-381) was subcloned into a modified pTRIEX2 vector with a C-terminal Flag-tag preceded by a TEV protease cleavage site (from here on termed TF). N-glycosylation sites were predicted and three potential sites, Asn114, Asn180, Asn352, were mutated via three mutations: Ser116Ala, Thr182Ala and Ser354Ala. This yielded PvCSS_22-381_S116A_T182A_S354A_TF. This construct was further cloned to

remove a predicted N-terminal repeat region to aid in crystallization. This yielded the construct PvCSS_115-381_S116A_T182A_S354A_TF. These constructs were expressed as soluble secreted proteins in HEK Expi293F cells (Life Technologies) as above.

To generate disulfide-linked PvPTRAMP-PvCSS (PvPC), PvPTRAMP and PvCSS were co-expressed in HEK Epi293F cells (Life Technologies) at a ratio of 50:50 PvPTRAMP:PvCSS for full length CSS and a ratio of 40:60 PvPTRAMP:PvCSS for PvCSS_115. The expression and purification were carried out as above with size-exclusion chromatography performed using either an S200 Increase 10/300 GL (Cytiva) or S200 16/600 HiLoad (Cytiva).

The full-length *pvripr* gene (PVP01_0816800, aa 22-1074) was subcloned into pAcGP67a with a C-terminal 6xHis-tag yielding PvRipr_22-1074_His and expressed in Sf21 cells using the flashBAC ULTRA baculovirus system (Oxford Expression Technologies). After initial transfection, the P1 virus was amplified and titrated several times to produce a P3 virus. This P3 virus was used for large scale expression. The proteins were expressed as soluble secreted proteins. The culture medium was harvested 3 days after the addition of P3 virus. The media was concentrated via tangential-flow filtration with a 3000 Da MWCO (Merck) to reduce the volume ~10-15 fold. This resultant concentrate was then dialyzed into TBS (20 mM Tris pH 8.5, 150 mM NaCl) overnight at 4 °C. Imidazole was then added to a final concentration of 10 mM and the culture media passed over Ni-NTA Agarose resin (Qiagen), washed with TBS + 20 mM imidazole, and eluted in TBS + 500 mM imidazole. The eluted protein was then concentrated on a 30,000 Da MWCO Amicon Ultra-15 Centrifugal Filter (Merck) and applied to an S200 Increase 10/300 GL column (Cytiva) connected to an Akta Pure (Cytiva). Peak fractions were pooled, concentrated and flash frozen as above. PvRipr truncations (PvRipr$^{tail}$(aa 665-1074), PvRipr$^{EGF\ 9,10,CTD}$(aa 844-1074), PvRipr$^{CTD}$(aa 972-1074)) were synthesized via the Genscript mutagenesis service. The truncated constructs were subcloned into the pAcGP67a vector and purified in an identical manner to the full-length construct.

PvRipr$^{EGF\ 6-8}$(aa 717-843) was synthesized and subcloned into pET28a yielding a construct with an N-terminal 6x His tag followed by a TEV site. Expression was carried out in *Escherichia coli* (*E. coli*) strain SHuffle® T7 (New England Biolabs) grown in terrific Broth with 40 μg/mL of kanamycin. One litre of culture was grown in incubators at 37 °C and shaking at 180 revolutions per minute (rpm) until an optical density at 600 nm (OD 600) of around 1.0 was reached. Isopropyl ß-D-1-thiogalactopyranoside (IPTG)(Astral) was then added to a final concentration of 1 mM, and protein expression was continued at 16 °C for 16-18 h. Cells were then harvested via centrifugation and the pellet resuspended in TBS pH 8.5, and with cOmplete ethylenediaminetetraacetic acid (EDTA)-free protease inhibitor cocktail (Roche). The resuspended cells were then sonicated, and the cellular extract clarified by centrifugation at 30,000 × *g* for 30 minutes at 4 °C. Imidazole was added to the clarified supernatant to a final concentration of 10 mM and then passed over pre-equilibrated Ni-NTA resin, washed with TBS pH 8.5 + 20 mM imidazole, and then eluted in TBS pH 8.5 containing 500 mM imidazole. The eluted protein was then concentrated on a 10,000 MWCO Amicon Ultra-15 Centrifugal Filter (Merck) and injected onto an S200 16/600 HiLoad (Cytiva) equilibrated in TBS pH 8.5. Peak fractions were then concentrated, supplemented with glycerol to a final concentration of 10% (v/v) and flash frozen in liquid nitrogen.

PvRipr$^{EGF\ 7-8}$ (764-843) was synthesized and subcloned into pcDNA34CK vector with the addition of a N-terminal 6xHis tag followed by a TEV site. Two predicted N-glycosylation sites, Asn767 and Asn780, were removed with the following mutations: Asn767Gln, Asn780Gln. This yielded the construct PvEGF 7-8$^{N767Q,\ N780Q}$. The protein was expressed in HEK Expi293F cells and purified in a similar manner to PvPTRAMP above. The proteins were purified via binding and eluting

using a 5 mL HisTrap Excel column (Cytiva) in PBS with 500 mM imidazole, followed by size-exclusion chromatography using an S200 Increase 10/300 column (Cytiva) connected to an Akta Pure (Cytiva) in TBS pH 8.0. Peak fractions were pooled and incubated with TEV protease overnight at 4 °C. His-tagged TEV was removed by applying the protein solution to Ni-NTA Agarose resin and collecting the flow-through. Protein was assessed for purity via SDS-PAGE. The flow-through was then buffer exchanged into TBS pH 8.0 and concentrated on a 3000 Da MWCO Amicon Ultra-4 Centrifugal Filter (Merck). Glycerol was added (10% v/v final) to the concentrated protein and then aliquoted and flash frozen in liquid $N_2$.

The *pvcyrpa* gene (PVP01_0532400, aa 24-362) was subcloned into pTRIEX2-TF vector which yielded PvCyRPA_22-362_TF. Two predicted N-glycoslyation sites, Asn78 and Asn282, were removed with the following mutations: Thr80Ala, Thr284Ala. The protein was expressed in HEK Expi293F cells and purified in an identical manner to PvpPTRAMP above.

PvRBP2b (PVX_094255, aa 161-1454) was purified as described previously[52].

### P. knowlesi
The *pkptramp* gene (PKNH_1437600, aa 21-297) excluding the transmembrane and cytoplasmic domains was subcloned into a modified pTRIEX2-SFT as was done for PvPTRAMP. Potential N-glycosylation sites were assessed and two sites, Asn115 and Asn261, were removed by mutation of Ser117 and Ser263 to Ala. This yielded SFT_PkPTRAMP_21-297_S117A_S263A. A PkPTRAMP construct expressing a C-terminal Avitag for biotinylation was made using PCR and restriction digests to yield SFT_PkPTRAMP_21-297_S117A_S263A-Avi. Expression of PkPTRAMP and PkPTRAMP-Avi was carried out in an identical manner to PvPTRAMP described above.

The *pkcss* gene (PKNH_1353400, aa 22-362) was subcloned into pTriEX2-TF. Five potential N-glycosylation sites, Asn96, Asn161, Asn175, Asn243 and Asn333 were removed via five mutations: Ser98-Ala, Thr163Ala, S177Ala, S245A and S335A. This yielded PkCSS_22-362_S98A_T163A_S177A_S245A_S335A _TF. Expression of PkCSS was carried out in an identical manner to PvCSS above.

Expression of the PkPC heterodimer was carried out in an identical manner as described for PvPC. The ratio of PkPTRAMP:PkCSS deoxyribonucleic acid (DNA) used was 60:40 when PkPTRAMP-SFT and PkCSS-TF were being used, and 50:50 when PkPTRAMP-Avi was being used.

The *pkripr* gene (PKNH_0817000, aa 22-1096) was subcloned into pAcGP67a with a C-terminal 6xHis-tag yielding PkRipr_22-1096_His, as per the PvRipr construct. Truncations of the full-length construct were made by Genscript using the mutagenesis service, to produce PkRipr^tail^(aa 669-1096), PkRipr^EGF 9,10,CTD^(aa 848-1096) and PkRipr^CTD^(aa 994-1096). All PkRipr constructs were expressed and purified in an identical manner to the equivalent PvRipr constructs.

For all constructs containing an Avitag, in vitro biotinylation was carried out as previously described[53].

### Antibodies and nanobodies
One alpaca was subcutaneously immunized six times 14 days apart with 130 µg (800 µg total) of recombinant PvPC. GERBU FAMA (GERBU Biotechnik GmbH, Heidelberg, Germany) was used as an adjuvant. Whole blood was collected three days after the last immunization for the preparation of lymphocytes. Nanobody library construction was carried out according to established methods[54]. Briefly, alpaca lymphocyte mRNA was extracted and amplified by reverse transcription PCR (RT-PCR) with nanobody-encoding, gene-specific primers. This produced a library of nanobody cDNA sequences that contained approximately $10^8$ sequences. The sequences that were cloned into the pMES4 phagemid vector were

amplified in *E. coli* TG1 strain and subsequently infected with M13KO7 helper phage for downstream recombinant phage expression. Handling of the alpaca for scientific purposes was approved by Agriculture Victoria, Wildlife and Small Institutions Animal Ethics Committee, project approval No. 26-17.

Biopanning was performed over two rounds with 1 µg of immobilized antigen as previously described[54]. Ninety-four positive clones were taken for further screening via enzyme-linked immunosorbent assay (ELISA). Clones showing positive binding by ELISA ($n = 93$) were sequenced. Of these, 71% were full length Variable Heavy domain of Heavy chain (VHH) ($n = 66$).

Nanobodies were expressed in the periplasm of *E. coli* WK6 cells as described previously[55]. Briefly, bacteria (250 mL) were grown in Terrific Broth at 37 °C to an OD 600 of 0.7. The cultures were then induced with 1 mM IPTG (Astral) and grown overnight at 28 °C. Cells were harvested and resuspended in PBS containing 20% sucrose and 20 mM imidazole to rupture the periplasm. EDTA was added to a final concentration of 5 mM, and the cells were incubated on ice. $MgCl_2$ was then added to a final concentration of 10 mM, and the periplasmic extract was harvested via centrifugation. The nanobodies were purified via standard Ni-NTA purification methods.

Monoclonal antibodies were raised in mice as per the Walter and Eliza Hall Animal Ethics Committee approved procedures. All monoclonal antibodies were produced by the WEHI Antibody Facility. Mice were injected with 80–180 µg of protein three times and then boosted once with 30–60 µg. After cloning of hybridomas, the supernatants were tested via ELISA and BLI. Based on these results, several hybridomas for each antigen were selected for further scale up of purified immunoglobulin G (IgG).

The plasmids of chimeric anti-PvRipr 5B3 Fab were synthesized by subcloning the variable domains of light and heavy chain of antibody 5B3 into pcDNA34CK and pcDNA34CH1 vectors respectively. The Fab was expressed in HEK Expi293F cells in a similar manner to PvPTRAMP above, with culture medium harvested 6 days post-transfection. The Fab was purified from the culture supernatant via binding and eluting using a 1 mL HiTrap MabSelect VL column (Cytiva) connected to an Akta Pure, in 50 mM sodium citrate buffer pH 2.5. The elution fractions are neutralized using 1 M Tris buffer pH 9.0, and the Fab was assessed for purity via SDS-PAGE. The elution fractions were pooled, concentrated and buffer exchanged into PBS on a 10,000 Da MWCO Amicon Ultra-15 Centrifugal Filter (Merck). The purified Fab was stored at 4 °C.

### Structure prediction
Prediction of PCR complexes from multiple *Plasmodium* species was done using the AlphaFold 3 server[34].

### Biolayer interferometry (BLI)
Biolayer interferometry (BLI) experiments were carried out on an Octet Red96e (Sartorius) at 25 °C. For kinetics analysis ligands were immobilized onto either anti-penta-His (His1K), streptavidin (SAX or SAX2) or Ni-NTA (NTA) biosensors (Sartorius) depending on the affinity tag present on the protein (His-tag or biotinylated Avitag). Ligands were diluted to 10–40 µg/mL in 1x kinetics buffer (PBS, pH 7.4, 0.1% (w/v) bovine serum albumin (BSA), 0.02% (v/v) Tween-20) prior to immobilization. Biosensors were initially dipped in kinetics buffer for 30–60 s to establish a baseline signal, and then dipped into wells containing the ligand, followed by another 30–60 s baseline. After the second baseline step, the ligands were then dipped into wells containing two-fold dilution series of analyte. Association was measured for 120 s and then the biosensors were dipped into kinetics buffer to measure the dissociation for another 120 s. Data were analysed using Sartorius Data Analysis software 11.0. Kinetic curves were fitted using a 1:1 binding model.

Competition studies for anti-PvPC nanobodies were performed using Ni-NTA (NTA) biosensors (Sartorius) with His-tagged nanobodies as the ligand (diluted to 5 µg/mL in kinetics buffer). After a 30 s baseline step, the biosensors were dipped into wells containing an irrelevant nanobody that does not bind to PvPC to quench the biosensor and ensure no free sites are present for the downstream steps. Following a second baseline step, the biosensors were dipped into PvPC diluted to 500 nM in kinetics buffer. After loading of PvPC onto the biosensors, a final baseline step was performed before the biosensors were dipped into either secondary nanobody (at 10 µg/mL diluted in kinetics buffer) or PvRipr (at 200 nM diluted in kinetics buffer). Data were analysed using Sartorius' Data Analysis software 11.0 and the epitope bins were assessed by normalization and manual curation.

Antibody kinetics were determined similarly to the above methods. Anti-Mouse IgG Fc Capture (AMC) biosensors (Sartorius) were used to immobilize mouse monoclonal antibodies at a concentration of 5–20 µg/mL in kinetics buffer. Antibody competition studies were carried out in a similar manner to the nanobodies, however anti-pentaHis (His1K) biosensors were used.

### Protein crystallization
PvPC_115 was purified as above. PvPC and nanobody D7 were co-complexed with the nanobody at 3x molar excess. The free nanobody was separated from the PvPC-nanobody complex by size-exclusion chromatography on an S200 Increase GL 10/300 (Cytiva) in HBS. Peak fractions were pooled and concentrated to ~5–6 mg/mL and set up in coarse screen sitting drop crystal trays at the Monash Macromolecular Crystallization Facility. Needle-like crystals formed after ~9 days in 0.2 M ammonium sulfate ($(NH_4)_2SO_4$) and 20% (w/v) polyethylene glycol (PEG) 3350. Further in-house optimization of conditions yielded large crystals in 0.2 M $(NH_4)_2SO_4$ and 16% (w/v) PEG-3350. Crystals were looped in mother liquor containing 10% (v/v) glycerol and flash frozen in liquid nitrogen. Diffraction data were collected with the MX2 beamline at the Australian Synchrotron (Clayton, Australia) at 100 K ($\lambda = 0.9537$ Å). Statistics are in Supplementary table 1.

PvEGF7-8$^{N767Q,N780Q}$ and 5B3 Fab were purified as above. PvEGF7-8$^{N767Q,N780Q}$ and 5B3 Fab were co-complexed with PvEGF7-8_N767Q_N780Q at 2x molar excess. Free PvEGF7-8_N767Q_N780Q was separated from the co-complex by size-exclusion chromatography on an S200 Increase GL 10/300 (Cytiva) in HBS pH 7.5. Fractions were assessed for purity and co-complexation via SDS-PAGE. Peak fractions were pooled and concentrated to 5 mg/mL and set up in coarse screen sitting drop crystal trays at the Bio21 Macromolecular Crystallization Facility. Thin-plate crystals formed after ~4 days in 0.2 M $(NH_4)_2SO_4$ and 25% w/v PEG 3350. Further in-house optimization of conditions yielded singular rod-like crystals in 0.2 M $(NH_4)_2SO_4$, 20% PEG-3,350 and 0.1 M sodium acetate buffer pH 4.4. Crystals were looped in mother liquor containing 15% (v/v) glycerol and flash frozen in $LN_2$. Diffraction data were collected with the MX2 beamline at the Australian Synchrotron (Clayton, Australia) at 100 K ($\lambda = 0.9537$ Å). Statistics are in Supplementary table 4.

### Structure determination and model building
Both diffraction data were processed with the XDS package[56] before being scaled and merged using Aimless[57] in the CCP4 suite[58]. The program Matthews[59] was used to estimate the number of molecules in the asymmetric unit.

For the structure of PvPC-D7 an AlphaFold 2[33] model of PvCSS constituting residues 115-381 was used as a search model for molecular replacement using Phaser[60]. After 3 copies of PvCSS were fitted, additional searches were performed with a nanobody structure. To ensure the best fit possible, a BLASTp (https://blast.ncbi.nlm.nih.gov/Blast.cgi?PAGE=Proteins) search was performed with nanobody D7

to find the structure with the highest sequence similarity for molecular replacement searches. This search yielded a nanobody (PDB: 7N0R), which was used as a search model with the complementarity determining region 3 (CDR3) sequence removed[61]. The structure was then iteratively refined in Phenix[62] and assessed and modified with Coot[63]. Clear density extended from the unpaired cysteine in PvCSS, C122, that was not accounted for by either PvCSS or nanobody D7. Due to the crystals being set up with the PvPC heterodimer, it was reasonable to conclude that this density belonged to PvPTRAMP. PvPTRAMP residues 41-53 were built into the electron density de novo independently for each of the three molecules in the asymmetric unit. Refinement and model statistics are described in Supplementary Table 1. For analysis of the contacts formed between PvPTRAMP and PvCSS, and PvCSS and D7, the program Contact (part of CCP4 suite)[58] was used in conjunction with PISA server (https://www.ebi.ac.uk/pdbe/pisa/) and are summarized in Supplementary Tables 2 and 3. Nanobodies were renumbered according to the Kabat numbering system, as determined by ANARCI[63].

For the structure of PvRipr EGF7-8 in complex with 5B3 Fab, 1 molecule was estimated in the asymmetric unit. The AlphaFold2 prediction of PvRipr EGF 7-8 and the crystal structure of an HIV-2 neutralizing Fab fragment (PDB: 3NZ8) with the CDRs deleted, were used as models for molecular replacement using Phaser. Alternating rounds of structure building and refinement were carried out using Coot and Phenix. Refinement and model statistics are found in Supplementary Table 4. Interactions between PvRipr EGF 7-8 and 5B3 were analyzed using PISA and CCP4 Contact and are summarized in Supplementary Table 5. 5B3 was renumbered according to the Kabat numbering system, as determined by ANARCI.

### Mutiple sequence alignment
Multiple sequence alignments were computed using ESPript 3.0[64].

### Mass photometry
Mass photometry experiments were carried out on a Two$^{MP}$ mass photometer (Refeyn). Each well was focused after the addition of 10 µL of filtered PBS. Once focused, 10 µL of protein at either 50 nM (*P. knowlesi*) or 100 nM (*P. vivax*) was added, mixed, and events were recorded for one minute using AcquireMP (Refeyn). Raw data processing was done in DiscoverMP (Refeyn) and the data exported and presented using Prism v9 (GraphPad). In-house recombinant mouse mixed lineage kinase-like (MLKL), human glutamine synthetase and human catalase were used for the construction of a calibration curve.

### Human samples
Human plasma samples were utilized from three cohorts of *Plasmodium* infected patients along with three cohorts of malaria-naïve negative controls. Patients infected with *P. vivax* were recruited from Tha Song Yang, Thailand, during the year 2014, as previously described[38], with a subset of 34 included in the current study. Patients infected with *P. falciparum*[37] and *P. knowlesi*[39,40] were recruited from Sabah, Malaysia, during the years 2010–2018 and 2012–2014, respectively. *P. falciparum* (*n* = 31) and *P. knowlesi* (*n* = 33) infected patients were included in the current study. Plasma samples were assayed from time of clinical presentation, 1 week later, and 1 month later. 28 malaria-naïve samples from the Melbourne Volunteer Biospecimen Donor Registry (VBDR) and 29 malaria-naïve samples from the Thai Red Cross (TRC) were utilized to create seropositivity cut-offs. Individuals from the TRC donated blood in Bangkok, a malaria-free region of Thailand, and had not had malaria diagnosed in the year prior nor had they traveled to endemic regions in the prior three years, as previously described[65]. An additional set of afebrile healthy controls (*n* = 30) were assayed from Sabah, Malaysia; however, these individuals may have

had prior *Plasmodium* infections and were thus not utilized to create the seropositivity cut-off.

Ethical approval for sample use was provided by WEHI Human Research Ethics Committee (14/02), with original study approval in Thailand (Faculty of Tropical Medicine, Mahidol University, MUTM 2014-025-01 and 02) and Malaysia (Menzies School of Health Research, HREC 12-1815, 16-2544, 10-1431, 12-1807). All individuals gave informed consent and/or assent to participate in the studies.

## Multiplexed antibody assay

Recombinant *Plasmodium* proteins were coupled to unique regions of magnetic, fluorescent, Bio-Plex microbeads (Bio-Rad) following the manufacturer's instructions and as previously described[66]. Briefly, 200 μL of microbeads were washed then activated for 20 min with sulfo-N-hydrosuccinimide (50 mg/mL) and N-ethyl-N-(3-dimethylaminopropyl) carbodiimide (EDC) (50 mg/mL) in monobasic sodium phosphate (pH 6.2). Following further washing, the activated microspheres were resuspended in PBS with 1–6 μg of *Plasmodium* protein. After overnight incubation, the coupled beads were washed and then stored in PBS-TBN (PBS, 0.1% (w/v) BSA, 0.02% (v/v) TWEEN-20, 0.05% (w/v) sodium azide, pH 7) at 4 °C until further use. Microbeads were always kept protected from light.

Plasma samples were diluted in PBT (1X PBS, 1% (w/v) BSA, 0.05% (v/v) Tween-20) at a dilution of 1:100. For *P. vivax* and *P. falciparum* antigens, plasma samples from hyper-immune individuals from PNG were used as a positive control. For *P. knowlesi* antigens, plasma samples from acutely infected *P. knowlesi* patients were used as the positive control. Both positive control pools were used to create a modified reference standard curve, starting at 1:50 with a 5 point 5-fold serial dilution. Diluted samples (50 μL) all controls and patients) were added to black flat-bottom 96-well plates and mixed with 50 μL of the coupled-antigen bead mixture (0.1 μL of each coupled antigen per well in PBT), then incubated for 30 minutes. The plate was washed and then 100 μL of 1:100 phycoerythrin (PE)-conjugated anti-human secondary antibody (Jackson Immunoresearch) was added and incubated for a further 15 min. Plates were washed then resuspended in PBT before being read on a MAGPIX instrument. Median fluorescent intensity was converted to arbitrary relative antibody units (RAU) using the standard curves, to adjust for plate-plate variation[65].

## Statistical analysis

An antigen-specific seropositivity cut-off was set as the mean of the negative controls (VBDR + TRC) plus two times the standard deviation. Data are presented as the fold change of the mean peak week 1 antibody response relative to the seropositivity cut-off. RAU values of samples and control cohorts are shown in Supplementary Fig. 10.

## Growth inhibition assays

*P. knowlesi* growth inhibition assays were undertaken using *P. knowlesi* YH1 parasites over 2 cycles of growth (~64 h) using standard conditions[67]. Antibodies were initially screened at 0.5 mg/mL for inhibitory activity before 2-fold serial dilution dose response curves were undertaken to define potency for inhibitory antibodies. Parasitemia was determined using flow cytometry (BD Acurri) after staining with 10 mg/mL of ethidium bromide, with data analysed using FlowJo software (BD Life Sciences). *P. knowlesi* growth in the presence of antibodies was compared to that of untreated control wells to define growth inhibitory activity. All experiments were performed a minimum of three times with duplicate wells unless stated otherwise.

*P. falciparum* growth inhibition assays were performed as described previously[18].

*P. vivax* growth was analysed using an ex vivo invasion assay performed as described previously with slight modification[68]. *P. vivax* samples were collected in 2023 from infected individuals in Kampong Speu, Western Cambodia. *P. vivax* infection was

determined using rapid diagnostic testing (CareStartTM Malaria Pf/pan rapid diagnostic tests, Accessbio) or microscopy and species-specific PCR to ensure monoinfection. Venous blood was collected in lithium heparin tubes and immediately sent on ice to the Malaria Research Unit at Institute Pasteur, Cambodia. There, erythrocytes were separated from the plasma, and the plasma was discarded. Erythrocytes were then suspended in warm Roswell Park Memorial Institute (RPMI) medium before leukocyte depletion using a non-woven fabric filter. The work presented here was approved by the National Ethics Committee for Health Research in Cambodia (192NECHR, July 11, 2022). All patients and/or their parents/guardians provided informed written consent for samples to be taken and used for these purposes.

Infected erythrocytes were enriched using a potassium chloride (KCl)-Percoll density gradient and then transferred into culture in supplemented Iscove's Modified Dulbecco's Medium (IMDM)(Gibco) (supplemented with 0.5% (w/v) Albumax II (Gibco), 2.5% (v/v) heat-inactivated human serum, 25 mM HEPES (Gibco), 20 μg/mL gentamicin (Sigma) and 0.2 mM hypoxanthine (C-C Pro)). The stage of the parasite culture was then assessed via thin blood smear. In the case of a majority ring culture, parasites were allowed to mature through to the schizont stage (~40 h) before starting the experiment. If the culture consisted mainly of trophozoites, the experiment was carried out after 18-20 h. The enriched schizonts were then mixed 1:1 with reticulocytes (previously enriched from cord blood or adult peripheral blood from malaria-naïve donors) and pre-labeled with Celltrace Far Red Dye for quantitation. The cultures were incubated with either 500 μg/mL (anti-tetanus toxin 43038) or either 100 μg/mL or 500 μg/mL (monoclonal antibodies and nanobodies) of biologics in a volume of 50 μL in 384-well plates. Cells were stained with Hoechst 33342 to stain parasite DNA and parasitemia was quantified via flow cytometry, with new infections being defined as Far Red/Hoechst double-positive cells. For quantitation, data were normalized against parasites mock treated with PBS. Observed control invasion rates ranged from 0.46 to 5.3% (median = 0.7%).

*P. cynomolgi* assays were performed as previously described[44]. Both *P. falciparum* and *P. knowlesi* GIAs were carried out in parallel to *P. cynomolgi* assays to serve as positive controls for antibody inhibition. *P. cynomolgi* strain Berok R9 was maintained in rhesus red blood cells (Emory Primate Center) at 2% hematocrit in RPMI 1640 with 10% human O+ serum and gassed (1% $O_2$, 5% $CO_2$, 94% $N_2$) at 37 °C. The invasion assay was set up with 0.2% hematocrit and 2–3% schizontemia in 30 μl volumes in 384-well plates using antibodies 2D9, 4H10, 5B3, 5B4, IgG control and heparin (positive control) After 12 h of incubation, the parasite DNA was stained with Vybrant™ DyeCycle™ Violet (Invitrogen), and 100,000 cells were analyzed via a Cytek-Northern Lights flow cytometer. Invasion was measured by the percentage of newly parasitized erythrocytes (CellTrace Far Red + / Vybrant™ DyeCycle™ Violet + ). Inhibition was assessed relative to control wells without antibodies. Data analysis was done using GraphPad Prism v10.

## Electron microscopy

All electron microscopy was carried out at the Bio21 Ian Holmes Imaging Centre, University of Melbourne.

For negative staining, purified PkPCR^tail + 5B3 Fab (at ~0.1 mg/mL) was applied to formvar and carbon coated, glow discharged copper grids (300 mesh, ProSciTech). Four microlitres of protein was incubated for one minute, then blotted off, washed twice in water, and then stained with 1% (w/v) uranyl acetate for two minutes before being blotted and dried thoroughly. The grids were then imaged on a Tecnai F30 operating at 200 kV. Two-dimensional classification was performed in cryoSPARC (v4.4.1)[69].

For cryo-electron microscopy, freshly purified PkPCR^tail and PkPCR^tail + 5B3 Fab at 0.5–1 mg/mL was applied to glow discharged

UltrAuFoil (Quantifoil Micro Tools GmbH) grids (300 mesh, R1.2/1.3) or HexAuFoil (Quantifoil Micro Tools GmbH) and then blotted for 5 seconds with a blot force of 7 (UltrAuFoil) or 10 (HexAuFoil) before being plunged into liquid ethane using a Vitrobot Mark IV (Thermo Fisher Scientific) operating at 4 °C and 100% humidity. Grids were screened for good ice quality on an FEI Talos Arctica (Thermo Fisher Scientific) operating at 200 kV. Grids showing sufficient thin and amorphous ice were then transferred to an FEI Titan Krios G4(Thermo Fisher Scientific) for data collection. Data were collected using an acceleration voltage of 300 kV and a Falcon 4i detector (Thermo Fisher Scientific) using EPU automation software. Data were collected over three sessions, two for PkPCR$^{tail}$ (from two independent grids) and one for PkPCR$^{tail}$ + 5B3. Pixel sizes used for collection were 0.506 Å/pixel with a total dose of 50 e⁻/Å$^2$ for PkPCR$^{tail}$ and 0.808 Å/pixel with a total dose of 40 e⁻/Å$^2$ for PkPCR$^{tail}$ + 5B3 and all datasets were collected with a nominal defocus range of −0.5 μm to −2 μm. CryoSPARC (v4.4.1-v4.6.2)[69] was used for all data processing. Gain and motion corrected, and contrast transfer function (CTF)-estimated movie stacks were curated to select for good CTF fit and to remove micrographs that showed signs of significant drift, contained obvious frost contamination, or had no visible particles. This resulted in 8730 and 826 movie stacks for session one and two, respectively, for PkPCR$^{tail}$ and 3285 movie stacks for PkPCR$^{tail}$ + 5B3. The curated datasets were then used in multiple rounds of automated picking and 2D class averaging. For PkPCR$^{tail}$ 11,955 particles corresponded to the 'front view', and 3067 particles corresponded to the 'side view'. For PkPCR$^{tail}$ + 5B3 13,478 particles corresponded to the 'front view', and 2694 particles corresponded to the 'side view'. Whilst clear features were visible in this small number of classes, severe orientation bias and the small, flat, and elongated shape of the particles precluded three-dimensional reconstruction.

### Reticulocyte enrichment for flow cytometric binding assays

Cord blood was obtained through a material transfer agreement (MTA, ID# M19/110) with the Bone Marrow Donor Institute (BMDI) at the Royal Children's Hospital in Melbourne, Australia under the human ethics project "14/09, Malaria parasite growth and invasion into reticulocytes" which was approved by the Walter and Eliza Hall Institute Human Research Ethics Committee (HREC). Cord blood was passed through a RC High Efficiency Leucocyte Removal Filter (Haemonetics Australia) and then centrifuged at 2000 x *g* for 5 min to separate blood from serum. The blood was then washed three times in 1x human tonicity PBS (HTPBS) before being made up to 50% hematocrit. This 50% solution was then layered on top of a 70% (v/v) Percoll cushion (GE Healthcare). Centrifugation for 25 min at 2100 x *g* separated the mature erythrocytes from the reticulocytes, with the reticulocytes forming a thin band at the interface between buffer and Percoll. Reticulocytes were stored in 1x HTPBS at 4 °C.

### Flow cytometry-based erythrocyte binding assays

For assays using mature erythrocytes, erythrocytes were washed twice in PBS and then made up to a density of approximately $1 \times 10^7$ cells/mL in PBS + 1% (w/v) BSA (PBS-BSA). Each sample used 100 μL of this suspension. Erythrocytes were centrifuged, the supernatant was removed, and the cells were resuspended in a solution containing freshly prepared recombinant proteins in PBS-BSA. Individual proteins were prepared at a final concentration of 2 μM (except for PfRh5, which was prepared at 400 nM), and complexes were mixed with an equimolar amount of protein to a final concentration of 2 μM. After a 45-min incubation at room temperature, the samples were centrifuged, washed, and then incubated with primary antibodies, either 5A9 (anti-Rh5), 4E2 (anti-PC) or 5E11 (anti-Ripr). After a 45-min incubation the cells were again centrifuged and then incubated with Alexa-488 anti-mouse fluorescent antibody at a dilution of 1:100. After a 45-min incubation, cells were washed twice in PBS and then resuspended

before analysis on an Attune NxT flow cytometer (Thermo Fisher Scientific). For each sample 50,000 events were recorded. The data were then analysed in FlowJo™ v10.7 Software (BD Life Sciences). Antibody background was subtracted from the positive population recorded in the presence of recombinant protein and this background-subtracted value has been plotted in the summary figures.

For assays involving reticulocyte-enriched cord blood, erythrocytes were made up in 1x HTPBS + 1% (w/v) BSA (HTPBS-BSA) to a density of approximately $1 \times 10^7$ cells/mL. Each sample used 100 μL of this suspension. Reticulocytes were centrifuged (2000 × *g* for one minute), the HTPBS-BSA removed, and then resuspended in a solution containing recombinant proteins in HTPBS-BSA and incubated at room temperature for 45 min. PvPC and PkPC were used at a final concentration of 2 μM. Samples were centrifuged after which the protein solution removed, and cells were washed once with HTPBS-BSA and then incubated with 4E2 (anti-PC) at a concentration of 0.05 mg/mL or polyclonal sera (anti-PvRBP2b) at a concentration of 12.5 μg/mL. After a 45-min incubation, the cells were again centrifuged, and the antibody solution was removed. Cells were washed once as before and then stained with Alexa-647 (either anti-rabbit or anti-mouse) at a dilution of 1:100. After 45 min the reticulocytes were again washed and incubated with 50 μL of thiazole orange (BD Retic-Count, BD Biosciences) for 30 min. Finally, the reticulocytes were centrifuged, the Retic-Count solution removed, and cells were washed with 1x HTPBS two times before analysis on an Attune NxT flow cytometer (Thermo Fisher Scientific). For each sample 50,000 events were recorded. The data were then analysed in FlowJo™ v10.7 Software (BD Life Sciences). This involved gating reticulocytes and then applying a quadrant gate according to the thiazole orange staining and the background staining of the antibody in combination with the Alexa 647. This antibody background was subtracted from the positive population recorded in the presence of recombinant protein: this background-subtracted value has been plotted in the summary figures. Positive binding is determined by the double positive population in the upper right-hand quadrant.

### Data visualization

All data visualization was done in University of California, San Francisco (UCSF) ChimeraX versions 1.2–1.8 (https://www.cgl.ucsf.edu/chimerax/)[70]. PyMOL was utilized for structure alignment and calculation of root mean square deviation (RMSD) (https://www.pymol.org/)[71].

### Reporting summary

Further information on research design is available in the Nature Portfolio Reporting Summary linked to this article.

## Data availability

The crystal structures generated in this study have been deposited in the Protein Data Bank, www.rcsb.org (PDB ID codes 9NSD and 9YIO). The underlying data and uncropped gels for all figures can be found in the Source Data file. Source data are provided with this paper.

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

## Acknowledgements

The authors thank Australian Red Cross Blood Service and Bone Marrow Donor Institute (BMDI) Cord Blood Bank for blood. We acknowledge Professor Jamie Rossjohn, the Monash Macromolecular Crystallization Facility (https://www.monash.edu/researchinfrastructure/mmcp), and the Bio21-WEHI Crystallization Facility within Melbourne Protein Characterization at The Bio21 Molecular Science and Biotechnology Institute, The University of Melbourne, where crystallization screening was undertaken. This research was undertaken in part using the MX2 beamline at the Australian Synchrotron, part of the Australian Nuclear Science and Technology Organisation, and made use of the Australian Cancer Research Foundation (ACRF) detector. We thank Professor Wai-Hong Tham for supply of the PvRBP2b plasmid and anti-PvRBP2b sera. We acknowledge all field teams in Thailand and Malaysia who contributed to collection of the used samples. We acknowledge the VBDR at WEHI for collection of Melbourne controls. This work was supported by the Gates Foundation (INV-074041), National Health and Medical Research Council of Australia (NHMRC) (grants 637406, APP1173049, GNT1173210), Drakensburg Trust, Australian Research Council (ARC FT240100420 University of Adelaide Research Scholarship), National Institutes of Health (NIH 5R01AI140751), and Victorian State Government Operational Infrastructure Support grant.

## Author contributions

B.A.S., P.S.L., X.X., and N.C.J. expressed proteins and performed and analyzed biophysical experiments. B.A.S. and X.X. solved crystal structures with assistance from S.W.S. B.A.S. analyzed cryo-EM data and wrote the initial draft manuscript. P.S.L., L.B.F.D., K.H.L., and S.D. performed and analyzed parasite growth inhibition assays. A.A. and P.S.L. performed and analyzed serological assays with supervision from R.J.L. T.W., M.J.G., N.M.A., J.S., and R.J.L. organized and collected patient plasma and clinical data for antibody analysis. A.L. carried out cryo-EM data collection. B.A.S., R.J.L., M.J.G., M.T.D., J.P., D.W.W., and S.W.S. designed and interpreted experiments. A.F.C. and S.W.S. designed and interpreted experiments and, along with B.A.S., wrote the final manuscript. All authors read and edited the manuscript.

## Competing interests

The authors declare no competing interests.
