## [Transparent Peer Review file · Nature Communications]

PTRAMP, CSS and Ripr form a conserved complex required for merozoite invasion of Plasmodium species into erythrocytes

Corresponding Author: Professor Alan Cowman

Version 0:

Reviewer comments:

Reviewer #1

(Remarks to the Author)

Key results: The manuscript provides an in-depth analysis of the formation and conservation of the heterotrimeric protein complex PC (PTRAMP-CSS) and RIPR in three Plasmodium species (Pf, Pv and Pk). The authors use advanced techniques such as predictive modelling, X-ray crystallography and cryo-EM to elucidate the complex's structural conformation, supported by biochemical and biophysical analyses that confirm protein interactions. While the complex exhibits notable structural conservation across species, species-specific functional and interaction differences are also identified, offering new insights into the molecular dynamics of erythrocyte invasion.

Validity: The structural and biophysical data interpretation is robust and well-supported by experimental results. However, certain aspects raise concerns regarding the validity of some conclusions:

- a. The discrepancy between the high structural conservation and the functional differences observed among species is not adequately addressed. This undermines the conclusion of analogous complex functionality across the three species. The discussion reveals a duality by initially suggesting similarity but also functional divergence. I recommend that the authors explore potential mechanisms that could explain this divergence in more depth.
- b. In inhibition assays, the requirement of high antibody concentrations to observe effects in *P. falciparum* raises uncertainty about whether inhibition is due to toxicity or a specific effect. Additional controls or a more detailed discussion are necessary.
- c. It is striking that antibody 5B3 exhibits cross-reactive inhibitory capacity in some species but fails to inhibit *P. vivax* invasion, which warrants further clarification from the authors.

Significance: This work addresses a topic of great importance in Plasmodium molecular biology, particularly regarding the mechanisms of cellular invasion, which are critical for the development of therapies and vaccines. The structural characterization of the PTRAMP-CSS-RIPR complex, along with interspecies functional differences, may open new research avenues and therapeutic strategies. Nevertheless, the lack of clarity in some interpretations might temporarily limit the study's impact.

Data and Methodology: The employed techniques (crystallography, Cryo-EM, BLI, SPR, biochemical assays) are appropriate and of high quality for the study's aims. The data presentation is clear, yet expansion of certain key experiments is advised:

- a. The absence of real-time studies on complex formation limits understanding of the dynamic assembly process, which could be improved by applying BLI or SPR techniques.
- b. It is also important to evaluate whether individual proteins interact directly with reticulocytes or if the complex must form prior to cell binding, as this has functional relevance.

Suggested Improvements

- a. Incorporate dynamic binding assays (real-time BLI or SPR) to clarify the assembly mechanism of the complex.
- b. Conduct additional studies to explain why, despite high structural conservation, functional differences exist among species, possibly investigating interactions with other proteins or the cellular environment.
- c. Explore the individual interaction of proteins with reticulocytes and potential interspecies functional differences.
- d. Clarify the potential toxic effects of high antibody concentrations in inhibition assays using appropriate controls.

Clarity and Context: The manuscript is written clearly and is accessible to specialists in the field. Results are adequately contextualized with previous studies, although some critical points, such as the functional discrepancies among species, require deeper discussion and justification.

This manuscript provides valuable insights into a protein complex relevant to Plasmodium molecular biology. With the suggested clarifications and additional experiments, the study could achieve the rigor and impact expected for publication in a high-profile journal. Therefore, I recommend consideration of the manuscript after a major that adequately addresses the points raised in this report.

Reviewer #2

(Remarks to the Author)

The manuscript by Seager and colleagues investigate the structural and epitope similarities between the PCR complex from *P. Knowlesi*, *P. falciparum* and *P. vivax*. It is a well written manuscript, that contains a large amount of data, both directly in the manuscript and as supplementary/extended data. The aim to identify cross-species reactive or even growth inhibitory antibody is very interesting and gives insight to both evolutionary adaptations of the different species as well as the possibilities to develop potential interventions/vaccines targeting all three species.

I find that the data support their conclusion well, the methods are sound and the data is not over-interpreted.

I though have a few comments and suggestion to corrections:

Line 137: the reference to figure 1d is a bit misleading. Fig. 1d rather shows, that the PC complex can also be seen with these two new non-glycosylated constructs, which could be written directly in the text. But it would also be good to show the glycosylated proteins in the same gel, so that the size differences caused by the glycosylations can be seen.

Line 141: The refence to Fig. 1e would be better to have in line 139, just after "KD of 160 nM" since it does not show the referenced data of affinity between CyRPA and RH5.

Regarding the PvRipr specific antibody, that 5B3, does it bind the PvPCR complex? I cannot see if it has. I think that information would support your discussion on why it does not inhibit the growth of *P. vivax*.

Fig. 5: When testing the antibodies in titration (f. 5d and e), you need to add the H2 (or another control antibody) in the same concentration span. This is especially relevant in fig. 5e where the concentration of 5B3 is being tested in really high concentrations.

Reviewer #3

(Remarks to the Author)

The authors have used AlphaFold3 to predict the structures of the PCR (PTRAMP, CSS and Ripr) complex from *Plasmodium falciparum*, *vivax* and *knowlesi*. Given that the proteins are conserved across *Plasmodium* species and that AF3 is trained on published structures, it is unsurprising that the AF3 prediction is highly similar for all three. The authors then crystallized the PvPC heterodimer, yielding a structure of PvCSS residues 115-381 and PvPTRAMP residues 42-53. The structure reveals a disulfide bond stabilizing the interaction between CSS and PTRAMP, which is conserved across the three species. Beyond this, the PvCSS structure is highly consistent with previously published PfCSS structures. The authors follow up with biophysical studies with recombinant protein to show that binding affinity of the various members for each other appear to be stronger than reported in previous studies using recombinant protein expressed in heterologous systems that differ from *Plasmodium* in terms of glycosylation. Finally, the authors show that anti-PCR antibodies are cross-reactive but show varying levels of growth inhibition across *Plasmodium* species. Overall, the work offers few technical or biological advances beyond published work and may be more suitable for a journal with a more focused readership.

1. Lines 260-276: The cryoEM 2Ds are sufficient for showing what general region of the complex a Fab or antibody binds to, but there is not enough information in low resolution 2D class averages to "strongly support the predicted PCR complex structure". Nor is the PvPC crystal structure, which contains only CSS residues 115-382 and PTRAMP residues 42-53, comprehensive enough to "strongly support the predicted PCR complex structure."

2. Have the authors attempted to obtain a 3D reconstruction from the 2D class averages shown in Fig. 6a?

3. The word "model" is used to variously refer to many different things throughout the manuscript, including a) atomic models from experimental crystal structures, b) predicted models from AlphaFold 3, c) models for biological or biophysical models d) cartoon models. This language should be clarified to avoid confusion. For example, the "models" in Fig 6c and on the right hand side of Fig 6b would more accurately be described as "cartoons" or "schematics" to differentiate from the predicted AF3

model on the left hand side of Fig 6b or the actual atomic model in Fig 2c.

Version 1:

Reviewer comments:

Reviewer #1

(Remarks to the Author)

Having reviewed the revised manuscript, the final report, and the point-by-point response letter, I am satisfied that the authors have made a clear and substantive effort to address the main concerns raised during peer review. In particular, the revisions strengthen the discussion of the structure–function relationship in the relevant protein complexes across the evaluated species, with a more transparent and coherent interpretation of the observations obtained using the 5B3 antibody.

I also appreciate the authors' commitment to pursuing additional crystallographic experiments aimed at elucidating underlying mechanisms and identifying conserved regions. These efforts will be valuable not only to reinforce the structural basis of the proposed functional interpretations, but also to better define which elements are truly conserved versus species-specific within this parasite system.

While some elements of the discussion necessarily remain interpretative and, at times, hypothesis-generating, they are generally framed appropriately and help delineate important open questions in these highly complex parasites. Importantly, the authors acknowledge that, although the species examined can infect similar host cell types, this does not imply conserved or identical molecular mechanisms, and the revised text reflects this nuance more carefully.

Overall, I consider that the manuscript has improved meaningfully and now provides a more balanced and rigorous framework for the conclusions.

Reviewer #2

(Remarks to the Author)

The authors have addressed all my comments made in the first review and I find the revised manuscript good and the results of significance to the field of malaria vaccine development.

Reviewer #3

(Remarks to the Author)

My previous concerns have been addressed. No further comments at this time.

Open Access This Peer Review File is licensed under a Creative Commons Attribution 4.0 International License, which permits use, sharing, adaptation, distribution and reproduction in any medium or format, as long as you give appropriate credit to the original author(s) and the source, provide a link to the Creative Commons license, and indicate if changes were

made.

Dear Editor

Many thanks for sending the reviews of our manuscript ‘PTRAMP, CSS and Ripr form a conserved complex required for merozoite invasion of *Plasmodium* species into erythrocytes’ by Seager et al. We have addressed the reviewers comments point by point below and made changes as suggested. Additionally, we have added substantially more data including a crystal structure determination of Ripr domains bound to the antibody 5B3.

We believe that this has addressed the reviewer’s comments and look forward to your final decision on this manuscript.

Kind regards
Alan Cowman

Author Response to Reviewers:

Reviewer #1 (Remarks to the Author):

Key results: The manuscript provides an in-depth analysis of the formation and conservation of the heterotrimeric protein complex PC (PTRAMP-CSS) and RIPR in three *Plasmodium* species (Pf, Pv and Pk). The authors use advanced techniques such as predictive modelling, X-ray crystallography and cryo-EM to elucidate the complex's structural conformation, supported by biochemical and biophysical analyses that confirm protein interactions. While the complex exhibits notable structural conservation across species, species-specific functional and interaction differences are also identified, offering new insights into the molecular dynamics of erythrocyte invasion.

Validity: The structural and biophysical data interpretation is robust and well-supported by experimental results. However, certain aspects raise concerns regarding the validity of some conclusions:

a. The discrepancy between the high structural conservation and the functional differences observed among species is not adequately addressed. This undermines the conclusion of analogous complex functionality across the three species. The discussion reveals a duality by initially suggesting similarity but also functional divergence. I recommend that the authors explore potential mechanisms that could explain this divergence in more depth.

Author response: We thank the reviewer for highlighting this important point. We acknowledge that our initial discussion may have not fully addressed the apparent discrepancy between structural conservation and functional differences observed among species. We have now expanded our discussion.

The high structural conservation of the PCR complex across *Plasmodium* species, as predicted by AlphaFold and supported by our structural and biochemical data, indeed suggests a conserved core function. However, our functional studies, particularly the differential inhibition by antibodies across species, reveal important species-specific differences.

To address this apparent discrepancy, we propose several potential mechanisms:

1. Subtle structural differences: While the overall structure is conserved, small differences in amino acid composition could lead to species-specific alterations in surface properties or dynamics of the complex.
2. Interaction with species-specific partners: The PCR complex likely interacts with other invasion proteins that are not conserved across species (such as PfRh5 in *P. falciparum*). These interactions could modulate the function or accessibility of the PCR complex in a species-specific manner.
3. Epitope accessibility: As suggested by our new structural data on the 5B3 antibody binding, the accessibility of certain epitopes may differ between species during the dynamic process of invasion, despite overall structural similarity.

We have included a more detailed discussion of these potential mechanisms in the manuscript (lines 379-406). While fully elucidating these differences will require further research, including structural studies of the complete invasion complexes from multiple species, we believe our work provides a crucial foundation for understanding both the conserved and divergent aspects of the PCR complex across Plasmodium species.

b. In inhibition assays, the requirement of high antibody concentrations to observe effects in *P. falciparum* raises uncertainty about whether inhibition is due to toxicity or a specific effect. Additional controls or a more detailed discussion are necessary.

Author Response: We have added a non-immune IgG control to the *P. falciparum* GIAs. The non-immune control data shows no growth defect in the presence of 10 mg/mL IgG, providing confidence that the results we see with the 5B3 mAb are specific. Further, non-inhibitory antibodies 2D9, 4E2 and 4H10 also show no effect on parasite growth at high concentrations. This additional data has been added to Main Figure 5 as follows:

Panel b updated with new competition for 4E2 and 2D9.

Panel c updated with new 4E2 inhibition

Panel d changed to include new reps and 4E2 data.

Panel e changed to include 4E2 mAb and non-immune IgG titrations up to 10 mg/ml.

c. It is striking that antibody 5B3 exhibits cross-reactive inhibitory capacity in some species but fails to inhibit *P. vivax* invasion, which warrants further clarification from the authors.

Author Response: We agree that the ability of 5B3 mAb to inhibit *P. falciparum* and *P. knowlesi* growth but not *P. vivax* or *P. cynomolgi* is a surprising result. To try to address this, we have solved the crystal structure of the 5B3 Fab in complex with EGF7 and EGF8 of PvRipr and defined the epitope. This additional data is described in a new Figure (Fig. 6). This figure includes a comparison of this epitope between PvRipr, PkRipr, PfRipr and PcRipr to show the strong homology of this region. Additionally, we determined the K_D of binding of 5B3 for PkRipr, PfRipr and PvRipr and whilst there are significant differences it does not explain the growth inhibition for Pf and Pk compared to Pv and Pc. These results suggest that subtle structural differences in the PfPCR and PkPCR complex, compared with that of PvPCR and PcPCR, may allow 5B3 access during merozoite invasion in the former species.

To fully define the difference that explains this result it would be necessary to solve the structure of the full complex with the Rh5 equivalent receptor binding protein attached for each with 5B3 bound using Cryo-EM. Currently, the Rh5 equivalent protein for Pk, Pv and Pc PCR

complexes are not identified. Whilst we are working towards this identification and plan to determine these structures using Cryo-EM this is a major effort that is outside the scope of this manuscript. However, we have added considerable additional data to start to address this question including a full figure that determines how 5B3 binds to Ripr.

Significance: This work addresses a topic of great importance in Plasmodium molecular biology, particularly regarding the mechanisms of cellular invasion, which are critical for the development of therapies and vaccines. The structural characterization of the PTRAMP-CSS-RIPR complex, along with interspecies functional differences, may open new research avenues and therapeutic strategies. Nevertheless, the lack of clarity in some interpretations might temporarily limit the study's impact.

Author Response: We appreciate the reviewer's recognition of the importance of our work. To address the concern about clarity in interpretations and to enhance the study's impact, we have made several substantial additions and clarifications:

1. **Structural characterization:** We have solved the crystal structure of the 5B3 Fab in complex with EGF7 and EGF8 of PvRipr, providing detailed structural information about this important epitope (new Main Figure 6). This adds to our understanding of the PTRAMP-CSS-RIPR complex structure across species.
2. **Interspecies functional differences:** We include a comparative analysis of the 5B3 epitope across PvRipr, PkRipr, PfRipr and PcRipr demonstrating strong homology.
3. Based on our new data, we propose that subtle structural differences in the PCR complex between species might affect the accessibility of the 5B3 epitope during merozoite invasion, potentially explaining the observed functional differences.
4. We have outlined the next steps needed to fully resolve remaining questions, including solving structures of the full complex with the Rh5 equivalent receptor binding protein for each species.

These additions provide a more comprehensive and nuanced understanding of the PTRAMP-CSS-RIPR complex across Plasmodium species. We believe this enhanced clarity strengthens the study's impact and opens up new avenues for research.

Data and Methodology: The employed techniques (crystallography, Cryo-EM, BLI, SPR, biochemical assays) are appropriate and of high quality for the study's aims. The data presentation is clear, yet expansion of certain key experiments is advised:

- a. The absence of real-time studies on complex formation limits understanding of the dynamic assembly process, which could be improved by applying BLI or SPR techniques.

Author Response: We agree that real time studies on complex formation can provide insight into the interactions required for complex formation. We have extensively used BLI with respect to PCRCR subunit interaction and it is included (Fig. 1E, Fig. 3 a-d, Fig. 4 b-d, Ext. Fig 3 b, Ext Fig. 4b, Ext Fig. 5 a-c)(Main Figures 1e, 3a-d, 4b-d, and Supplementary Figures 7b, 8b, and 9a-c in the resubmitted manuscript). We have shown how Ripr engages the PTRAMP-CSS heterodimer with high affinity via the C-terminal domain and EGF domains 9 and 10. All BLI experiments were carried out over the same time of 240 seconds, with association being measured over the course of 120 seconds. This shows that complex formation is quite rapid.

To further illustrate the dynamic assembly process, we direct the reviewer to Figure 3 and 4, which shows representative BLI sensorgrams for the association of Ripr with the PTRAMP-CSS heterodimer. The rapid association observed (reaching equilibrium within 120 seconds) suggests that complex formation is not likely to be a rate-limiting step in the invasion process. This rapid assembly is consistent with the need for quick deployment of invasion machinery during the brief window of merozoite viability outside the red blood cell.

b. It is also important to evaluate whether individual proteins interact directly with reticulocytes or if the complex must form prior to cell binding, as this has functional relevance. Author response: We agree with the reviewer that understanding how this complex interacts with either erythrocytes and/or reticulocytes is crucial to its role in invasion and that is why we did the experiments originally presented in Extended Data. To this end, we carried out reticulocyte and normocyte binding experiments with both the *P. vivax* PC heterodimer and the *P. knowlesi* PC heterodimer, and erythrocyte binding experiments with the same proteins as well as the *P. knowlesi* PCR complex including the positive controls ie. PfRh5 and PvRBP2b (originally shown in **Extended Data figure 10**). These experiments showed that these complexes could not bind to reticulocytes or normocytes. It has also been shown previously that the *P. falciparum* PC heterodimer is unable to bind to erythrocytes without incorporation into the PCRCR complex (Scally et al. Nature Micro 2022). Performing the equivalent experiment with PvPCR was not feasible due to the low yields of recombinant full length PvRipr and the high concentrations required for the assay. We believe these findings support the idea that PTRAMP-CSS-Ripr complex is a scaffold for other species-specific red blood cell binding proteins (such as PfRh5 in *P. falciparum*). Whilst the binding data to reticulocytes and normocytes is negative, we do believe it is an important piece of data, given that previous experiments with PfPTRAMP (Siddiqui et al Cell Microbiol 2013) and PkPTRAMP (Knuepfer et al. PLOS Pathogens 2019) suggested that PTRAMP was able to bind erythrocytes. We believe that the use of heterodimeric PTRAMP-CSS, rather than monomeric PTRAMP, provides more relevant data on how these proteins interact with host cells.

To make these results clearer and more visible we have moved the reticulocyte and normocyte binding assays to the new Figure 7 panel c and d.

Suggested Improvements

a. Incorporate dynamic binding assays (real-time BLI or SPR) to clarify the assembly mechanism of the complex.

Author response: We have included BLI experiments that address the assembly of this complex. PTRAMP and CSS are present as a heterodimer before binding to the Ripr CTD. This complex (along with other potential interactors) is present within the micronemes before being exposed on the merozoite surface. Further cell biology and/or imaging experiments will hopefully shed further light on this complex, but we believe such experiments are outside the scope of this study.

b. Conduct additional studies to explain why, despite high structural conservation, functional differences exist among species, possibly investigating interactions with other proteins or the cellular environment.

Author Response: We appreciate the reviewer's suggestion to further investigate the reasons for functional differences despite high structural conservation among species. We have conducted additional studies and analyses to address this important point:

1. Structural analysis of antibody binding: We have determined the crystal structure of the 5B3 Fab in complex with PvRipr EGF7-8 (new Figure 6). This structure provides insight into the specific epitope recognized by this cross-reactive but differentially inhibitory antibody.
2. Comparative epitope analysis: We included a comparison of the 5B3 epitope across PvRipr, PkRipr, PfRipr and PcRipr. While this region shows strong homology, we identified subtle differences that may contribute to functional divergence.
3. Cellular environment considerations: Our reticulocyte and normocyte binding assays (now in Figure 7, panels c and d) suggest that the PCR complex alone is not sufficient for host cell binding in *P. vivax* and *P. knowlesi*, unlike the situation in *P. falciparum* where the PCRCR complex can bind erythrocytes.

Based on these new data, we propose that functional differences may arise from:

1. Subtle structural variations affecting epitope accessibility during the dynamic process of invasion.
2. Differences in how the PCR complex interacts with other, potentially species-specific, invasion proteins.
3. Variations in the cellular context of invasion, such as the preference for reticulocytes in *P. vivax*.

We acknowledge that fully explaining these functional differences will require additional research, including:

1. Identification and characterization of species-specific binding partners of the PCR complex.
2. Structural studies of the complete invasion complexes from multiple species.
3. Dynamic studies of complex assembly and function during the invasion process.

While these extensive studies are beyond the scope of the current work, our findings provide a crucial foundation for understanding both the conserved and divergent aspects of the PCR complex across *Plasmodium* species. We have expanded our discussion in the manuscript to address these points and outline future research directions.

c. Explore the individual interaction of proteins with reticulocytes and potential interspecies functional differences.

Author Response: As noted above, we carried out erythrocyte and reticulocyte binding assays. Both of these experiments showed no binding for PC heterodimers or the PkPCR trimeric complex. As we have shown that both PTRAMP and CSS exist as a disulfide-linked heterodimer, we believe that assessing the binding of monomeric proteins would not reflect how these proteins function in the parasite. Performing the equivalent experiment with PvPCR was not feasible due to the low yields of recombinant full length PvRipr and the high concentrations required for the assay.

Given our previous findings that the complete PCRCR complex is required for PTRAMP-CSS erythrocyte binding in *P. falciparum*, we believe that other, yet unidentified, proteins are required for PCR binding in *P. knowlesi* and *P. vivax*.

These results have now been incorporated into figure 7 (panels c and d).

d. Clarify the potential toxic effects of high antibody concentrations in inhibition assays using appropriate controls.

Author Response: We have now added a non-immune IgG control to the *P. falciparum* GIAs. The new control data shows no growth defect in the presence of 10mg/mL IgG, giving us confidence that the results we see with 5B3 are specific. Further, non-inhibitory antibodies 2D9, 4E2 and 4H10 also show no effect on parasite growth at high concentrations.

Clarity and Context: The manuscript is written clearly and is accessible to specialists in the field. Results are adequately contextualized with previous studies, although some critical points, such as the functional discrepancies among species, require deeper discussion and justification.

Author Response: See comments above.

This manuscript provides valuable insights into a protein complex relevant to *Plasmodium* molecular biology. With the suggested clarifications and additional experiments, the study could achieve the rigor and impact expected for publication in a high-profile journal. Therefore, I recommend consideration of the manuscript after a major that adequately addresses the points raised in this report.

Reviewer #2 (Remarks to the Author):

The manuscript by Seager and colleagues investigate the structural and epitope similarities between the PCR complex from *P. Knowlesi*, *P. falciparum* and *P. vivax*. It is a well written manuscript, that contains a large amount of data, both directly in the manuscript and as supplementary/extended data. The aim to identify cross-species reactive or even growth inhibitory antibody is very interesting and gives insight to both evolutionary adaptations of the different species as well as the possibilities to develop potential interventions/vaccines targeting all three species.

I find that the data support their conclusion well, the methods are sound and the data is not over-interpreted.

I though have a few comments and suggestion to corrections:

Line 137: the reference to figure 1d is a bit misleading. Fig. 1d rather shows, that the PC complex can also be seen with these two new non-glycosylated constructs, which could be written directly in the text. But it would also be good to show the glycosylated proteins in the same gel, so that the size differences caused by the glycosylations can be seen.

Author Response: We have now included the glycosylated PfPC gel in figure 1 panel d as requested.

Line 141: The refence to Fig. 1e would be better to have in line 139, just after “KD of 160 nM” since it does not show the referenced data of affinity between CyRPA and RH5.

Author Response: We have moved the reference as per the reviewer’s request.

Regarding the PvRipr specific antibody, that 5B3, does it bind the PvPCR complex? I cannot see if it has. I think that information would support your discussion on why it does not inhibit the growth of *P. vivax*.

Author Response: Yes, 5B3 can directly bind the PvPCR complex and this is shown in Fig 5c and Extended Figure 8c (now Supplementary Figure 12 in the resubmitted manuscript) which

show the competition epitope binning results. PvPC is still able to bind PvRipr once 5B3 is bound, indicating that a PvPCR-5B3 complex is possible. We agree if 5B3 could not bind the complex but bound PvRipr this would explain the inhibition of growth studies. We have inserted two sentences in the Discussion that discusses the 5B3 results. Lines 379-381:

“Interestingly, the Ripr-binding mAb 5B3, raised against PvRipr, exhibited an unexpected pattern of cross-inhibitory activity. It inhibited invasion of *P. knowlesi* and *P. falciparum*, but not *P. vivax* or *P. cynomolgi*, despite binding to the PvPCR complex.”

Fig. 5: When testing the antibodies in titration (f. 5d and e), you need to add the H2 (or another control antibody) in the same concentration span. This is especially relevant in fig. 5e where the concentration of 5B3 is being tested in really high concentrations.

Author Response: As mentioned above we have now added a non-immune IgG control to the *P. falciparum* GIAs. The new control data shows no growth defect in the presence of 10 mg/mL IgG, giving us confidence that the results we see with 5B3 are specific. Further, non-inhibitory antibodies 2D9, 4E2 and 4H10 also show no effect on parasite growth at high concentrations.

Reviewer #3 (Remarks to the Author):

The authors have used AlphaFold3 to predict the structures of the PCR (PTRAMP, CSS and Ripr) complex from *Plasmodium falciparum*, *vivax* and *knowlesi*. Given that the proteins are conserved across *Plasmodium* species and that AF3 is trained on published structures, it is unsurprising that the AF3 prediction is highly similar for all three. The authors then crystallized the PvPC heterodimer, yielding a structure of PvCSS residues 115-381 and PvPTRAMP residues 42-53. The structure reveals a disulfide bond stabilizing the interaction between CSS and PTRAMP, which is conserved across the three species. Beyond this, the PvCSS structure is highly consistent with previously published PfCSS structures. The authors follow up with biophysical studies with recombinant protein to show that binding affinity of the various members for each other appear to be stronger than reported in previous studies using recombinant protein expressed in heterologous systems that differ from *Plasmodium* in terms of glycosylation. Finally, the authors show that anti-PCR antibodies are cross-reactive but show varying levels of growth inhibition across *Plasmodium* species. Overall, the work offers few technical or biological advances beyond published work and may be more suitable for a journal with a more focused readership.

Authors Response: We appreciate the reviewer's summary of our work. However, we believe our study offers significant technical and biological advances beyond published work. We would like to highlight the key findings and emphasize the additional data we have added in response to reviewer comments:

Key Findings and Advances:

1. Conservation of PCR complex: We demonstrate that the PTRAMP-CSS-Ripr (PCR) complex is conserved across all *Plasmodium* species, including those that lack Rh5, suggesting a fundamental role in invasion beyond the *P. falciparum*-specific PCRCR complex.
2. Structural insights: Our crystal structure of the PvPC heterodimer provides the first structural evidence of the PTRAMP-CSS interaction outside of *P. falciparum*, confirming the conservation of this critical disulfide bond.
3. Improved biophysical characterization: By optimizing protein expression to reduce glycosylation, we demonstrate significantly stronger binding affinities between

complex components than previously reported, providing a more accurate representation of these interactions.

4. Species-specific functional differences: Despite structural conservation, we reveal species-specific differences in antibody-mediated growth inhibition, highlighting important functional divergences in the PCR complex across *Plasmodium* species.
5. Cross-reactive antibodies: We identify antibodies that show cross-reactivity and differential inhibition across multiple *Plasmodium* species, which has significant implications for vaccine development.

Additional Data and Analyses:

In response to reviewer comments, we have added substantial new data and analyses:

1. Crystal structure of 5B3 Fab-PvRipr complex: We determined the structure of the inhibitory antibody 5B3 in complex with PvRipr EGF7-8, providing detailed structural information about this important epitope (new Figure 6).
2. Comparative epitope analysis: We performed a detailed comparison of the 5B3 epitope across PvRipr, PkRipr, PfRipr and PcRipr revealing subtle differences that may contribute to functional divergence.
3. Additional growth inhibition controls: We added non-immune IgG controls and data for non-inhibitory antibodies at high concentrations to demonstrate the specificity of inhibitory effects.
4. Improved presentation of binding assays: We moved the reticulocyte and normocyte binding assay results to the main figures (Figure 7c, d) for greater visibility and impact.

These findings and additional data provide significant new insights into the conservation, structure, and function of the PCR complex across *Plasmodium* species. They reveal important species-specific differences that have implications for understanding invasion mechanisms and developing cross-species interventions. We believe this work represents a substantial advance in our understanding of *Plasmodium* invasion biology and has broad relevance to the field of malaria research.

1. Lines 260-276: The cryoEM 2Ds are sufficient for showing what general region of the complex a Fab or antibody binds to, but there is not enough information in low resolution 2D class averages to “strongly support the predicted PCR complex structure”. Nor is the PvPC crystal structure, which contains only CSS residues 115-382 and PTRAMP residues 42-53, comprehensive enough to “strongly support the predicted PCR complex structure.”

Author Response: We acknowledge that the 2D class averages from cryo-EM and our partial crystal structure of PvPC alone are not sufficient to definitively prove the entire PCR complex structure. However, we believe that the combination of these structural data with our extensive biochemical and biophysical analyses provides strong support for our proposed model of protein interactions within the complex.

The crystal structure of PvCSS with PvPTRAMP residues 42-53 confirms the presence and location of the intermolecular disulfide bond predicted by AlphaFold. This key interaction forms the basis of the PC heterodimer, which our biochemical data show is crucial for high-affinity Ripr binding.

Our new crystal structure of the 5B3 Fab with PvRipr EGF7-8 provides additional structural information about a key region of Ripr involved in complex formation. When combined with

our binding studies showing that the C-terminal region of Ripr is sufficient for PC binding, this helps to constrain the possible arrangements of Ripr within the complex.

The cryo-EM 2D class averages, while low resolution, are consistent with our proposed model and provide visual confirmation of the overall shape and dimensions of the complex. Importantly, they show the 5B3 Fab binding to the predicted location on Ripr.

While we agree that no single piece of our data conclusively proves the entire complex structure, we believe that the consistency across multiple independent lines of evidence - structural, biochemical, and biophysical - provides strong support for our proposed model of the PCR complex.

2. Have the authors attempted to obtain a 3D reconstruction from the 2D class averages shown in Fig. 6a?

Author response: We thank the reviewer for the suggestion. Indeed, we tried extensively to generate a reasonable 3D reconstruction for the PCR complex. Due to the orientation bias, we were unable to produce a 3D reconstruction of sufficient quality for publication. Shown below is an ab-initio reconstruction of PkPCR at low resolution, showing that the model fits reasonably well within the density. We were unable to improve the resolution further and the density shown here clearly suffers from anisotropy.

This figure shows ab-initio cryoEM density (grey) and AlphaFold model of PkPCR (purple for PTRAMP, green for CSS and yellow for Ripr)(loops have been omitted for clarity).

3. The word “model” is used to variously refer to many different things throughout the manuscript, including a) atomic models from experimental crystal structures, b) predicted models from AlphaFold 3, c) models for biological or biophysical models d) cartoon models. This language should be clarified to avoid confusion. For example, the “models” in Fig 6c and on the right hand side of Fig 6b would more accurately be described as “cartoons” or “schematics” to differentiate from the predicted AF3 model on the left hand side of Fig 6b or the actual atomic model in Fig 2c.

Authors Response: We thank the reviewer for bringing this to our attention. Where possible we have tried to qualify the use of model by including a specifier (e.g. predicted model). We have revised our use of the model throughout the manuscript to minimize confusion.

Many thanks for considering our manuscript. The reviewer's comments have been very helpful and improved the manuscript and we are pleased they are now happy with our additional data and changes. All of the editorial changes and formatting required have been done and attached.

We look forward to your finale decision

.